# A Free Lunch from the Noise:
## Provable and Practical Exploration for Representation Learning

## Abstract

Representation learning lies at the heart of the empirical success of deep learning for dealing with the curse of dimensionality. However, the power of representation learning has not been fully exploited yet in reinforcement learning (RL), due to **i)**, the trade-off between expressiveness and tractability; and **ii)**, the coupling between exploration and representation learning. In this paper, we first reveal the fact that under some noise assumption in the stochastic control model, we can obtain the linear spectral feature of its corresponding Markov transition operator in closed-form *for free*. Based on this observation, we propose *Spectral Dynamics Embedding (SPEDE)*, which breaks the trade-off and completes optimistic exploration for representation learning by exploiting the structure of the noise. We provide rigorous theoretical analysis of SPEDE, and demonstrate the practical superior performance over the existing state-of-the-art empirical algorithms on several benchmarks.

## 1 Introduction

Reinforcement learning (RL) dedicates to solve the sequential decision making problem, where an agent is interacting with an *unknown* environment to find the best policy that maximizes the expected cumulative rewards (Sutton & Barto, 2018). It is known that the tabular algorithms direct controlling over the original state and action achieve the minimax-optimal regret depending on the cardinality of the state and action space (Jaksch et al., 2010; Osband & Van Roy, 2016; Azar et al., 2017; Jin et al., 2018). However, these algorithms become intractable for the real-world problems with an enormous number of states, due to the curse of dimensionality. Learning with function approximation upon *good* representation is a natural idea to tackle the curse and serving as the key for the success of deep learning (Bengio et al., 2013). In fact, representation learning lies at the heart of the empirical successes of deep RL in video games (Mnih et al., 2013), robotics (Levine et al., 2016), Go (Silver et al., 2017), dialogue systems (Jiang et al., 2021) to name a few. Meanwhile, the importance and benefits of the representation in RL is rigorously justified (Jin et al., 2020; Yang & Wang, 2020), which quantifies the regret in terms of the dimension of the *known* representation based on a subclass in Markov decision processes (MDPs) (Puterman, 2014). A natural question raises:

> *How to design* **provably efficient** *and* **practical** *algorithm for representation learning in RL?*

Here, by "provably efficient" we mean the sample complexity of the algorithm can be rigorously characterized only in terms of the complexity of representation class, without explicit dependency on the number of states and actions, while by "practical" we mean the algorithm can be implemented and deployed for the real-world applications. Therefore, we not only require the representation learned is expressive enough for handling complex practical environments, but also require the operations in the algorithm tractable and computation/memory efficient. The major difficulty of this question lies in two-fold:

    **i)** The *trade-off* between the expressiveness and the tractability in the design of the representations;

    **ii)** The learning of representation is intimately *coupled* with exploration.

Specifically, a desired representation should be sufficiently expressive[1] to capture the dynamic system, while still tractable in practice. However, in general, expressive representation leads to complicated optimization in learning. For example, the representation in the linear MDP is *exponential*

---

[1]For a formal definition of expressiveness, see (Agarwal et al., 2020a).

*stronger* than the latent variable MDPs in terms of expressiveness (Agarwal et al., 2020a). However, its representation learning is either depending on a MLE oracle that is computational intractable due to the constraint on the regularity of conditional density (Agarwal et al., 2020a), or a complicated constrained min-max-min-max optimization (Modi et al., 2021). On the other hand, Misra et al. (2020) considers the representation introduced by an encoder in block MDP (Du et al., 2019), in which the learning problem can be completed by a regression, but with the payoff that the representations in block MDP is even weaker than latent variable MDP (Agarwal et al., 2020a).

Meanwhile, the coupling of the learning of representation and exploration induces the difficulty in *practical* algorithm design and analysis. One cannot learn a precise representation without enough experiences from a comprehensive exploration, while the exploration depends on an reliable estimation of the representation. Most of the known results depends on a policy-cover-based exploration (Du et al., 2019; Misra et al., 2020; Agarwal et al., 2020a; Modi et al., 2021), which maintains and samples a set of policies during training for systematic exploration, that significant increases the computation and memory cost in implementation.

In this work, we propose Spectral Dynamics Embedding (SPEDE), bypassing the aforementioned difficulties and answering the question affirmatively. SPEDE is established on an observation that connects the stochastic control dynamics (Osband & Van Roy, 2014; Kakade et al., 2020) with linear MDPs in Section 3. Specifically, by exploiting the property of the *noise* in the stochastic control dynamics, we can recover the factorization of its corresponding Markov transition operator in closed-form *without extra computation*. This equivalency immediately overcome the computational intractability in the model estimation in Agarwal et al. (2020a), which breaks the trade-off between expressiveness and tractability.

More importantly, the connection unifies two highly-related but commonly-known different models, *i.e.*, stochastic nonlinear control models (Kakade et al., 2020) and linear MDPs (Jin et al., 2020), therefore, provides the opportunity to share benefits from both sides: **i)**, it sheds the light on exploiting optimistic control for exploratory representation learning, instead of expensive policy-cover-based exploration; and **ii)**, it introduces the linear sufficient feature from the spectral space of Markov operator, in which the planning can be completed efficiently.

We rigorously characterize the statistical property of SPEDE in terms of regret w.r.t. the complexity of representation class in Section 4, without explicit dependence on the raw feature of state and action. With the established unified view, our results generalize online control (Kakade et al., 2020) and linear MDP (Jin et al., 2020) beyond *known* features. We finally demonstrate the superior of SPEDE on the MuJoCo benchmarks in Section 5. It significantly outperforms the empirical state-of-the-art RL algorithms. To our knowledge, SPEDE is the first representation learning algorithm achieving statistical, computational, and memory efficiency with sufficient expressiveness.

## 1.1 RELATED WORK

There have been many great attempts to learn a variety of **algorithmic representation learning** in RL designed for different purposes, *e.g.*, bisimulation (Ferns et al., 2004; Gelada et al., 2019; Castro, 2020), reconstruction (Watter et al., 2015; Hafner et al., 2019). Recently, there are also several works considering the spectral features based on decomposing different variants of the transition operator, including successor features (Dayan, 1993; Kulkarni et al., 2016), proto-value functions (Mahadevan & Maggioni, 2007; Wu et al., 2018), spectral state-aggregation (Duan et al., 2018; Zhang & Wang, 2019), and contrastive fourier features (Nachum & Yang, 2021). These works are highly-related to the proposed SPEDE. Besides these features focus on *state-only* representation, the major difference between SPEDE and these spectral features lies in **i)**, the target operators in existing spectral features are *state-state* transition, which cancel the effect of action; **ii)**, the target operators are estimated based on empirical data from a *fixed behavior policy* under the implicit assumption that the estimated operator is *uniformly accurate*, ignoring the major difficulty in exploration, while SPEDE carefully designed the systematic exploration with theoretical guarantee; **iii)**, most of the existing spectral features rely on *explicitly* decomposition of the operators, while SPEDE obtains the spectral *for free*.

Turning to the **rigorously-justified representation learning**, a large body of effort focuses on policy-cover-based methods in an "explore-then-commit" strategy (Du et al., 2019; Misra et al., 2020; Agarwal et al., 2020a; Modi et al., 2021). These algorithms learn a uniformly accurate model/representation through a reward-free exploration, upon which decouple the learning from the exploration. Then, the representation from the learned model can be used for optimal policy seeking for the particular reward. The major difficulty impedes their practical application is the computa-

tion and memory cost: the policy-cover-based exploration requires a set of exploratory polices to be maintained and sampled from during training, which can be extremely expensive.

Another two related lines of research are **model-based RL** and **online control**, which are commonly known overlapped but separate communities considering different formulations of the dynamics. Our finding establishes the equivalency among the model, so that bridges these communities. Osband & Van Roy (2014) and Kakade et al. (2020) are the most related to our work in each community. These models generalize their corresponded linear models, *i.e.*, Jin et al. (2020) and Cohen et al. (2019), with general nonlinear model and kernel function within a known RKHS, respectively. The regret of the optimstic (pessimistic) algorithm has been carefully characterized for these models. However, both of the proposed algorithms in Osband & Van Roy (2014) and Kakade et al. (2020) require an planning oracle to seek the optimal policy, which might be computational intractable. In SPEDE, this is easily handled in the equivalent linear MDP.

## 2 PRELIMINARIES

Markov Decision Process (MDP) is one of the most standard models studied in the reinforcement learning that can be denoted by the tuple $\mathcal{M} = (\mathcal{S}, \mathcal{A}, r, T, \mu, H)$, where $\mathcal{S}$ is the state space, $\mathcal{A}$ is the action space, $r : \mathcal{S} \times \mathcal{A} \to \mathbb{R}^+$ is the reward function[2] (where $\mathbb{R}^+$ denotes the set of non-negative real numbers), $T : \mathcal{S} \times \mathcal{A} \to \Delta(\mathcal{S})$ is the transition and $\mu$ is an initial state distribution and $H$ is the horizon[3] (i.e. the length of each episode). A (potentially non-stationary) policy $\pi$ can be defined as $\{\pi_h\}_{h \in [H]}$ where $\pi_h : \mathcal{S} \to \mathcal{A}, \forall h \in [H]$. Following the standard notation, we define the value function $V_h^\pi(s_h) := \mathbb{E}_{T,\pi}\left[\sum_{t=h}^{H-1} r(s_t, a_t) | s_h = s\right]$ and the action-value function (i.e. the $Q$ function) $Q_h^\pi(s_h, a_h) = \mathbb{E}_{T,\pi}\left[\sum_{t=h}^{H-1} r(s_t, a_t) | s_h = s, a_h = a\right]$, which are the expected cumulative rewards under transition $T$ when executing policy $\pi$ starting from $s_h$ and $(s_h, a_h)$. With these two definitions at hand, it is straightforward to show the following Bellman equation:
$$Q_h^\pi(s_h, a_h) = r(s_h, a_h) + \mathbb{E}_{s_{h+1} \sim T(\cdot | s_h, a_h)} V_{h+1}^\pi(s_{h+1}).$$
Reinforcement learning aims at finding the optimal policy $\pi^* = \arg\max_\pi \mathbb{E}_{s \sim \mu} V_0^\pi(s)$. It is well known that in the tabular setting when the state space and action space are finite, we can provably identify the optimal policy with both sample-efficient and computational-efficient optimism-based methods (*e.g.* Azar et al., 2017; Jin et al., 2018; Zhang et al., 2021) with the complexity proportion to $|\mathcal{S}||\mathcal{A}|$. However, in practice, the cardinality of state and action space can be large or even infinite. Hence, we need to incorporate function approximation into the learning algorithm when we deal with such cases. The linear MDP (Jin et al., 2020) or low-rank MDP (Agarwal et al., 2020a; Modi et al., 2021) is the most well-known reinforcement learning model that can incorporate linear function approximation with theoretical guarantee, thanks to the following assumption on the transition:
$$T(s'|s, a) = \langle \phi(s, a), \mu(s') \rangle_{\mathcal{H}}, \tag{1}$$
where $\phi : \mathcal{S} \times \mathcal{A} \to \mathcal{H}, \mu : \mathcal{S} \to \mathcal{H}$ are two feature maps and $\mathcal{H}$ is a Hilbert space. The most essential observation for them is that, $Q_h^\pi(s, a)$ for any policy $\pi$ is linear w.r.t $\phi(s_h, a_h)$, due to the following observation (Jin et al., 2020):
$$\int V_{h+1}^\pi(s_{h+1}) T(s_{h+1}|s_h, a_h) \, \mathrm{d}s_{h+1} = \left\langle \phi(s_h, a_h), \int V_{h+1}^\pi(s_{h+1}) \mu(s_{h+1}) \, \mathrm{d}s_{h+1} \right\rangle_{\mathcal{H}}. \tag{2}$$
Therefore, $\phi$ serves as a sufficient representation for the estimation of $Q_h^\pi$, that can provide uncertainty estimation with standard linear model analysis and eventually lead to sample-efficient learning when $\phi$ is fixed and known to the agent (see Theorem 3.1 in Jin et al., 2020). However, we in general do not have such representation in advance[4] and we need to learn the representation from the data, which constraints the applicability of the algorithms derived with fixed and known representation.

## 3 SPECTRAL DYNAMICS EMBEDDING

It is naturally to consider how to perform sample-efficient representation learning (and hence sample-efficient reinforcement learning) that satisfies (1) in an online manner. The most straight-

---

[2]In general, the reward can be stochastic. Here for simplicity we assume the reward is deterministic and known throughout the paper, which is a common assumption in the literature (*e.g.*, Jin et al., 2018; 2020; Kakade et al., 2020).

[3]Our method can be generalized to infinite horizon case, see Section 3.2 for the detail.

[4]One exception is the tabular MDP, where we can choose $\phi : \mathcal{S} \times \mathcal{A} \to \mathbb{R}^{|\mathcal{S}|^2|\mathcal{A}|}$ that each state-action pair has exclusive $|\mathcal{S}|$ non-zero element and $\mu : \mathcal{S} \to \mathbb{R}^{|\mathcal{S}|^2|\mathcal{A}|}$ correspondingly defined to make (1) hold.

forward idea is performing the maximum likelihood estimation (MLE) in the representation space (*e.g.*, Agarwal et al., 2020a). Unfortunately, for general cases, such MLE is intractable, due to the constraints on the regularity of marginal distribution (*i.e.*, $\langle \phi(s,a), \int_{s'} \mu(s')\,\mathrm{d}s' \rangle = 1$) for all $(s,a) \in \mathcal{S} \times \mathcal{A}$. Moreover, even we can perform MLE for certain cases (for example, the block MDP), as the representation is estimated from the data, which can be inaccurate, all of the existing work apply the policy cover technique (Du et al., 2019; Misra et al., 2020; Agarwal et al., 2020a; Modi et al., 2021) to enforce exploration. However, such procedure can be both computational and memory expansive when we need amounts of exploratory policy to guarantee the coverage of whole state space, which makes it not a good practical choice.

To overcome these issues, we introduce Spectral Dynamics Embedding (SPEDE), which leverages the noise structure to provide a simple but provable efficient and practical algorithm for representation learning in RL. We first introduce our key observation, which provides us a practical way to implement the optimism in the face of uncertainty (OFU) principle.

### 3.1 KEY OBSERVATION

Our most important observation is that, the density of isotropic Gaussian distribution can be expressed as the inner product of two feature maps, thanks to the reproducing property and the random Fourier transform of the Gaussian kernel[5] (Rahimi & Recht, 2007):

$$\phi(x|\mu, \sigma^2 I) \propto \exp\left(-\frac{\|x - \mu\|^2}{2\sigma^2}\right)$$

$$\textit{(Reproducing Property)} = \langle k(x,\cdot), k(\mu,\cdot) \rangle_{\mathcal{H}} \tag{3}$$

$$\textit{(Random Fourier Transform)} = \langle \varphi(x,\omega,b), \varphi(\mu,\omega,b) \rangle_{p(\omega,b)}, \tag{4}$$

where $k(\cdot,\cdot)$ is the Gaussian kernel with bandwith $\sigma$: $k(x,y) = \exp\left(-\frac{\|x-y\|_2^2}{2\sigma^2}\right)$, $\mathcal{H}$ is the Reproducing Kernel Hilbert Space (RKHS) associated with $k$, $\varphi(x,\omega,b) = \sqrt{2}\cos(\omega^\top x + b)$, $\langle f,g \rangle_p = \mathbb{E}_{p(x)}[f(x)g(x)]$ and $p(\omega,b) = \mathcal{N}(\omega; 0, 1/\sigma^2 I) \cdot \mathcal{U}(b; [0,2\pi])$ with $\mathcal{N}$ and $\mathcal{U}$ denoting Gaussian and Uniform distribution, respectively.

Consider the general transition dynamics,

$$s' = f^*(s,a) + \epsilon, \epsilon \sim \mathcal{N}(0,\sigma^2), \quad \text{equivalently} \quad T(s'|s,a) \propto \exp\left(-\frac{\|s' - f^*(s,a)\|^2}{2\sigma^2}\right), \tag{5}$$

which is the basic setup in the empirical model-based reinforcement learning (*e.g.*, Chua et al., 2018; Kurutach et al., 2018; Clavera et al., 2018; Wang et al., 2019), and the online (non)-linear control (*e.g.*, Abbasi-Yadkori & Szepesvári, 2011; Mania et al., 2019; 2020; Simchowitz & Foster, 2020; Kakade et al., 2020). Here $s \in \mathbb{R}^d$, $a \in \mathcal{A}$ that can be continuous and $f^*$ is a dynamic function. By applying the reproducing property (3) or random Fourier transform (4) for the transition dynamics (5), we can obtain the feature $\phi$ and $\mu$ satisfies (1) *for free*. Specifically, taking the reproducing property as an example, we have that

$$T(s'|s,a) = \langle k(f^*(s,a),\cdot), (2\pi\sigma^2)^{-d/2} k(s',\cdot) \rangle_{\mathcal{H}}, \tag{6}$$

which means the problem (5) is indeed a linear MDP with $\phi(s,a) = k(f^*(s,a),\cdot)$ and $\mu(s') = (2\pi\sigma^2)^{-d/2} k(s',\cdot)$. Following (2), we know $Q(s,a)$ is in the linear span of the $\phi(s,a)$ that is transformed from $f^*(s,a)$. Therefore, finding a good representation of $(s,a)$ is equivalent to finding a good estimation of $f^*$. In the next section, we will show that, with the well-known optimism in the face of uncertainty (OFU) principle, we can estimate $f^*$ in an online manner with a both sample-efficient and computational-efficient algorithm.

**Remark (Computation-free Factorizable Noise Model):** We want to remark that, similar observations also hold for large amounts of distribution, *e.g.*, the Laplace and Cauchy distribution. We refer the interested reader to Table 1 in Dai et al. (2014) for the known transformation of kernels and features. Here we focus on the Gaussian noise, as Gaussian distribution is the most natural continuous distribution we should work on.

### 3.2 PRACTICAL ALGORITHM DESCRIPTION

Here, we introduce a generic Thompson Sampling (TS) type algorithm in Algorithm 1 based on the OFU principle that leverage our observation at the previous section. At the beginning, we provide a

---

[5]We provide a brief review on the related definitions in Appendix A.

---

**Algorithm 1** Thompson Sampling (TS) Algorithm

---

**Require:** Number of Episodes $K$, Prior Distribution $\mathbb{P}(f)$, Reward Function $r(s,a)$.
 1: Initialize the history set $\mathcal{H}_0 = \emptyset$.
 2: **for** episodes $k = 1, 2, \cdots$ **do**
 3:     Sample $f_k \sim \mathbb{P}(f|\mathcal{H}_k)$.                            ▷ Draw the Representation.
 4:     Find the optimal policy $\pi_k$ on $f_k$ with Algorithm 2.        ▷ Planning with $f_k$.
 5:     **for** steps $h = 0, 1, \cdots, H-1$ **do**                  ▷ Executing $\pi_k$.
 6:         Execute $a_h^k \sim \pi_k^h(s_h^k)$.
 7:         Observe $s_{h+1}$.
 8:     **end for**
 9:     Set $\mathcal{H}_k = \mathcal{H}_{k-1} \cup \{(s_h^k, a_h^k, s_{h+1}^k)\}_{h=0}^{H-1}$.        ▷ Update the History.
10: **end for**

---

**Algorithm 2** Planning with Dynamic Programming

---

**Require:** Transition Model $f$, Reward Function $r(s,a)$.
 1: Initialize $\phi(s,a)$, $\mu(s')$ with (3) or (4). $V_H(s) = 0, \forall s$.
 2: **for** steps $h = H-1, H-2, \cdots, 0$ **do**
 3:     Calculate $Q_h(s,a) = r(s,a) + \langle \phi(s,a), \int V_{h+1}(s')\mu(s')\,\mathrm{d}s' \rangle_{\mathcal{H}}$.      ▷ Bellman Update.
 4:     Set $V_h(s) = \max_a Q_h(s,a)$, $\pi_h(s) = \arg\max_a Q_h(s,a)$.      ▷ Choose the Optimal Policy.
 5: **end for**
 6: **return** $\{\pi_h\}_{h=0}^{H-1}$.

---

prior distribution $\mathbb{P}(f)$ that reflects our prior knowledge on $f^*$. Then for each episode, we draw a $f$ from the posterior, find the optimal policy with $f$ using the planning algorithm, execute this policy and eventually inference the posterior with the new observation. Notice that, we choose the policy optimistically with an *sampled* $f$, which enforces the exploration following the principle of OFU. Meanwhile, we only learn the dynamic with posterior inference and directly obtain the representation with Equation (3) or (4), which avoids additional error from the representation learning step. As all of our data is collected with $f^*$, our posterior will shrink to a point mass of $f^*$, which guarantees we can identify good representation and good policy with sufficient number of data.

One significant part of SPEDE is the computational-efficient planning with $f_k$, thanks to the linear MDP formulation (6). Prior work assumes an oracle (*e.g.*, Kakade et al., 2020) for such planning problem, but little is known on how to provably perform such planning efficiently. Notce that, with the feature $\phi(s,a)$ defined via (3) and (4), we know that $Q_h^\pi(s,a)$ is exactly linear in $\phi(s,a)$, $\forall h, \pi$. Hence, we can perform a dynamic programming style algorithm that calculate $Q_h(s,a)$ with the given feature $\phi(s,a)$, and then greedily select the action at each level $h$, which is simple yet efficient. It is straightforward to show that the policy obtained with this dynamic programming algorithm is optimal with the mathematical induction. We illustrate the detailed algorithm in Algorithm 2.

In such planning algorithm, we need to calculate the term $\int V_{h+1}(s')\mu(s')\,\mathrm{d}s'$ and take the maximum of $Q_h(s,a)$ over $a$, which can be problematic when the number of states and actions can be large or even infinite. We will provide more discussion on this issue later.

We then address several practical issues in implementing the proposed SPEDE:

**Posterior Sampling** The exact posterior inference can be hard if $f^*$ doesn't lie in simple function class (*e.g.*, linear function class) or has some derived property (*e.g.*, conjugacy), so in practice we apply the existing mature approximate inference methods like Markov Chain Monte Carlo (MCMC) (*e.g.*, Neal et al., 2011) and variational inference (*e.g.*, Blei et al., 2017; Kingma & Welling, 2013). In our implementation, we used Stochastic Gradient Langevin Dynamics (Welling & Teh, 2011; Cheng & Bartlett, 2018) to train an ensemble of models for achieving the purpose.

**Large State and Action Space** In general, we need to handle the case when the number of states and actions can be large, or even infinite. Notice that, when the state space is large, we can estimate the term $\int V_{h+1}(s')\mu(s')\,\mathrm{d}s'$ with regression based method using the samples from $f$ (Ernst et al., 2005; Antos et al., 2008). For the continuous action space, we can apply principled policy optimization methods (*e.g.*, Agarwal et al., 2020b) with an energy-based model (EBM) parametrized policy (Nachum et al., 2017; Dai et al., 2018), treat the linear $Q^\pi(s,a)$ as the gradient and perform mirror descent and eventually obtain the optimal policy. However, this is at the cost of an additional sampling step from the EBM policy. In practice, we introduce a Gaussian policy and perform soft actor-critic (Haarnoja et al., 2018) policy update, which already provides good empirical perfor-

mance. To sum up, for large state and action cases, we learn the critic in the learned representation space by regression, and obtain the Gaussian parametrized actor with SAC policy update step, in Line 3 and 4 in Algorithm 2, respectively.

**Infinite Horizon Case** Our algorithm can be provably extended to the infinite horizon case with specific termination condition for each episode (*e.g.*, see Jaksch et al., 2010). And in practice, for the planning part we can solve the linear fixed-point equation with the feature $\phi(s, a)$ using the popular algorithms like Fitted $Q$-iteration (FQI) (Ernst et al., 2005; Antos et al., 2008). that still guarantees to find the optimal policy.

## 4 THEORETICAL GUARANTEES

In this section, we provide theoretical results for SPEDE, showing that SPEDE can identify informative representation and as a result, near-optimal policy in a sample-efficient way. We first define the notation of regret. Assume at episode $k$, the learner chooses the policy $\pi_k$ and observes a sequence $\{(s_h^k, a_h^k)\}_{h=0}^{H-1}$. We define the regret of the first $K$ episodes (and define $T := KH$) as:

$$\text{Regret}(K) := \sum_{k \in [K]} \left[ V_0^*(s_0^k) - V_0^{\pi_k}(s_0^k) \right] \tag{7}$$

We want to provide a regret upper bound that is sublinear in $T$, as when $T$ increases, we collect more data that can help us build a much more accurate estimation on the representation, which should decrease the per-step regret and make the overall regret scale sublinear in $T$. As we consider the Thompson Sampling algorithm, we would like to study the expected regret $\mathbb{E}_{\mathbb{P}(f)}[\text{Regret}(K)]$, which takes the prior $\mathbb{P}(f)$ into account.

### 4.1 ASSUMPTIONS

Before we start, we first state the assumptions we use to derive our theoretical results.

**Assumptions on the environment** We make the following assumption on the environment, which is common in the literature (*e.g.* Azar et al., 2017; Jin et al., 2018; 2020).

**Assumption 1** (Bounded Reward). $r(s, a) \in [0, 1]$, $\forall (s, a) \in \mathcal{S} \times \mathcal{A}$.

**Assumptions on the function class $\mathcal{F}$** In practice, we generally approximate $f^*$ with some complicated function approximators, so we focus on the setting where we want to find $f^*$ from a general function class $\mathcal{F}$ that need not to be linear with certain feature map. This can be helpful for MuJoCo benchmarks that have angle, angular velocity and torque of the agent in the raw state, which we don't know how to construct the feature map to make $f$ linear. We first state some necessary definitions and assumptions on $\mathcal{F}$.

**Definition 1** ($\ell_2$-norm of functions). *Define* $\|f\|_2 := \max_{(s,a) \in \mathcal{S} \times \mathcal{A}} \|f(s, a)\|_2$. *Notice that it is not the commonly used $\ell_2$ norm for the function, but it suits our purpose well.*

**Assumption 2** (Bounded Output). *We assume that* $\|f\|_2 \leq C$, $\forall f \in \mathcal{F}$.

**Assumption 3** (Realizability). *We assume the ground truth dynamic function* $f^* \in \mathcal{F}$.

We then define the notion of covering number, which will be helpful in our algorithm derivation.

**Definition 2** (Covering Number (Wainwright, 2019)). *An $\epsilon$-cover of $\mathcal{F}$ with respect to a metric $\rho$ is a set $\{f_i\}_{i \in [n]} \subseteq \mathcal{F}$, such that $\forall f \in \mathcal{F}$, there exists $i \in [n]$, $\rho(f, f_i) \leq \epsilon$. The $\epsilon$-covering number is the cardinality of the smallest $\epsilon$-cover, denoted as $\mathcal{N}(\mathcal{F}, \epsilon, \rho)$.*

**Assumption 4** (Bounded Covering Number). *We assume that* $\mathcal{N}(\mathcal{F}, \epsilon, \|\cdot\|_2) < \infty, \forall \epsilon > 0$.

**Remark** Basically, Assumption 2 means the the transition dynamic never pushes the state far from the origin, which holds widely in practice. Assumption 3 guarantees that we can find the exact $f^*$ in $\mathcal{F}$, or we will always suffer from the error induced by model mismatch. Assumption 4 ensures that we can estimate $f^*$ with small error when we have sufficient number of observations.

Besides the bounded covering number, we also need an additional assumption on bounded eluder dimension, which is defined in the following:

**Definition 3** ($\epsilon$-dependency (Osband & Van Roy, 2014)). *A state-action pair $(s, a) \in \mathcal{S} \times \mathcal{A}$ is $\epsilon$-dependent on $\{(s_i, a_i)\}_{i \in [n]} \subseteq \mathcal{S} \times \mathcal{A}$ with respect to $\mathcal{F}$, if $\forall f, \tilde{f} \in \mathcal{F}$ satisfying $\sqrt{\sum_{i \in [n]} \|f(s_i, a_i) - \tilde{f}(s_i, a_i)\|_2^2} \leq \epsilon$ satisfies that $\|f(s, a) - \tilde{f}(s, a)\|_2 \leq \epsilon$. Furthermore, $(s, a)$ is said to be $\epsilon$-independent of $\{(s_i, a_i)\}_{i \in [n]}$ with respect to $\mathcal{F}$ if it is not $\epsilon$-dependent on $\{(s_i, a_i)\}_{i \in [n]}$.*

**Definition 4** (Eluder Dimension (Osband & Van Roy, 2014)). *We define the eluder dimension $\dim_E(\mathcal{F}, \epsilon)$ as the length $d$ of the longest sequence of elements in $\mathcal{S} \times \mathcal{A}$, such that $\exists \epsilon' \geq \epsilon$, every element is $\epsilon'$-independent of its predecessors.*

**Remark** Intuitively, eluder dimension illustrates the number of samples we need to make our prediction on unseen data accurate. If the eluder dimension is unbounded, then we cannot make any meaningful prediction on unseen data, even we have large amounts of samples. Hence, to make the learning possible, we need the following bounded eluder dimension assumption.

**Assumption 5** (Bounded Eluder Dimension). *We assume* $\dim_E(\mathcal{F}, \epsilon) < \infty, \forall \epsilon > 0$.

### 4.2 MAIN RESULT

**Theorem 5** (Regret Bound). *Assume Assumption 2 to 5 holds. We have that*

$$\mathbb{E}_{\mathbb{P}(f)}\left[\text{Regret}(K)\right] \leq \tilde{O}(\sqrt{H^2 T \cdot \log \mathcal{N}(\mathcal{F}, T^{-1/2}, \|\cdot\|_2) \cdot \dim_E(\mathcal{F}, T^{-1/2})}).$$

*where $\tilde{O}$ represents the order up to logarithm factors.*

For finite dimensional function class, $\log \mathcal{N}(\mathcal{F}, T^{-1/2}, \|\cdot\|_2)$ and $\dim_E(F, T^{-1/2}))$ should be scaled like $\text{polylog}(T)$, hence our upper bound is sublinear in $T$. The proof is in Appendix C. Here we briefly sketch the proof idea.

*Proof Sketch.* We first construct an equivalent UCB algorithm (see Appendix B) and bound $\text{Regret}(K)$ for it. Then by the conclusion from Russo & Van Roy (2013; 2014); Osband & Van Roy (2014), we can directly translate the upper bound on $\text{Regret}(K)$ from UCB algorithm to an upper bound on $\mathbb{E}_{\mathbb{P}(f)}[\text{Regret}(K)]$ of TS algorithm.

With the optimism, we know for episode $k$, $V_0^*(s_0^k) \leq \tilde{V}_{0,k}^{\pi_k}(s_0^k)$, where $\tilde{V}_{h,k}^{\pi_k}$ is the value function of policy $\pi_k$ under the model $\tilde{f}_k$ introduced in the UCB algorithm. Hence, the regret at episode $k$ can be bounded by $\tilde{V}_{0,k}^{\pi_k}(s_0^k) - V_0^{\pi_k}(s_0^k)$, which is the value difference of the policy $\pi_k$ under the two models $\tilde{f}_k$ and $f^*$, that can be bounded by $\sqrt{\mathbb{E}\left[\sum_{h=0}^{H-1} \|f^*(s_h^k, a_h^k) - \tilde{f}_k(s_h^k, a_h^k)\|_2^2\right]}$ (see Lemma 14 for the details), which means when the estimated model $\hat{f}$ is close to the real model $f^*$, the policy obtained by planning on $\hat{f}$ will only suffer from a small regret. With Cauchy-Schwartz inequality, we only need to bound $\mathbb{E}\left[\sum_{k \in [K]} \sum_{h=0}^{H-1} \|f^*(s_h^k, a_h^k) - \tilde{f}_k(s_h^k, a_h^k)\|_2^2\right]$. This term can be handled via Lemma 18. With some additional technical steps, we can obtain the upper bound on $\text{Regret}(K)$ for the UCB algorithm, and hence the upper bound on $\mathbb{E}_{\mathbb{P}(f)}[\text{Regret}(K)]$ for the TS algorithm. $\square$

**Kernelized Non-linear Regulator** Notice that, for the linear function class $\mathcal{F} = \{\theta^\top \varphi(s, a) : \theta \in \mathbb{R}^{d_\varphi \times d}\}$ where $\varphi : \mathcal{S} \times \mathcal{A} \to \mathbb{R}^{d_\varphi}$ is a fixed and known feature map of certain RKHS[6], when the feature and the parameters are bounded, the logarithm covering number can be bounded by $\log \mathcal{N}(\mathcal{F}, \epsilon, \|\cdot\|_2) \lesssim d_\varphi \log(1/\epsilon)$, and the eluder dimension can be bounded by $\dim_E(\mathcal{F}, \epsilon) \lesssim d_\varphi \log(1/\epsilon)$ (see Appendix D for the detail, notice that we provide a tighter bound of the eluder dimension compared with the one derived in Osband & Van Roy (2014)). Hence, for linear function class, Theorem 5 can be translated into a regret upper bound of $\tilde{O}(H d_\varphi T^{1/2})$ for sufficient large $T$, that matches the results of Kakade et al. (2020)[7]. Moreover, for the case of linear bandits when $H = 1$, our bound can be translated into a regret upper bound of $\tilde{O}(d_\varphi T^{1/2})$, that matches the lower bound (Dani et al., 2008) up to logarithmic terms.

**Compared with Kakade et al. (2020) and Osband & Van Roy (2014)** Our results have some connections with the results from Kakade et al. (2020) and Osband & Van Roy (2014). However, in Kakade et al. (2020), the authors only considers the case when $\mathcal{F}$ only contains linear functions w.r.t some known feature map, which constrains its application in practice. We instead, consider the general function approximation, which makes our algorithm applicable for more complicated models like deep neural networks. Meanwhile, the regret bound from Osband & Van Roy (2014) depends on a global Lipschitz constant for the value function, which can be hard to quantify with either theoretical or empirical method. Instead, our regret bound gets rid of such dependency on the Lipschitz constant with the simulation lemma that carefully exploit the noise structure.

---

[6]Note that, the RKHS here is the Hilbert space that contains $f(s, a)$ with the feature from some fixed and known kernel, It is different from the RKHS we introduced in Section 3, that contains $Q(s, a)$ with the feature $k(f(s, a), \cdot)$ where $k$ is the Gaussian kernel.

[7]Note that $T$ in (Kakade et al., 2020) is the number of episodes, and $V_{\max}$ in (Kakade et al., 2020) can be viewed as $H^2$ when the per-step reward is bounded.

## 5 EXPERIMENTS

In this section, we study the empirical performance of SPEDE in the OpenAI MuJoCo control suite (Brockman et al., 2016). We use the environments from MBBL (Wang et al., 2019), which varies slightly from the original environments in terms of modifying the reward function so its gradient w.r.t. the states exists and introducing early termination (ET). Note that the set of environments contains various control and manipulation tasks, which are commonly used for benchmarking both model-free and model-based RL algorithms (Yu et al., 2020; Kakade et al., 2020; Luo et al., 2018; Haarnoja et al., 2018). As aforementioned, for practical implementation, our critic network consists of a representation network $\phi(\cdot)$ and a linear layer on the top. We follow the same procedure of Algorithm 1: (1) For finding the optimal policy, we run an actor-critic algorithm (SAC). (2) We update the representation network of the critic function $\phi(\cdot)$ with a momentum from the model dynamics network $f(\cdot)$. We found this prevent updating the representation of too fast. We provide the full set of experiments in Appendix E.2, an ablation on the momentum in Appendix E.3 and the hyperparameter we use in Appendix E.6. [8]

**Baselines** We compare our method with various model-based RL baselines: PETS (Chua et al., 2018) with random shooting (RS) optimizer, PETS with cross entropy method (CEM) optimizer and ME-TRPO (Kurutach et al., 2018). Note that these are strong empirical baselines with many hand-tuned hyperparameters and engineering features (*e.g.*, ensemble of models). It is usually hard for any theoretically guaranteed model-based RL algorithm to match or surpass their performance (Kakade et al., 2020). Another natural baseline is the successor feature (Dayan, 1993), which is one of the representative spectral features. We compare with the deep successor feature (DeepSF) (Kulkarni et al., 2016), and for a fair comparison, we only swap the representation objective of SPEDE with DeepSF and keep the other parts of the algorithm exactly the same.

**SPEDE: Performance with the Learned Representation** Following Algorithm 1, we are interested in how SPEDE performs when we conduct planning on top of the representation induced by the dynamics model in each episode. As most of the rigorously-justified representation learning algorithms are computationally intractable/inefficient, to demonstrate the effectiveness of representation used in SPEDE, we compare SPEDE with the deep successor features, which is one representative empirical representation learning algorithm. Moreover, as our method essentially models transition dynamics, we compare our methods with the state-of-the-art model-based RL algorithms. We summarize the results of our method in Table 2. We see that our method achieves impressive performance comparing to model-based RL methods. Even in some hard environments that baselines fail to reach positive reward (*e.g.*, MountainCar, Walker-ET), SPEDE manage to achieve a reward of 52.6 and 501.6 respectively. We also evaluate our representation by comparing SPEDE to the usage of deep successor feature (DeepSF). Results show that on hard tasks like Humanoid and Walker, SPEDE manages to achieve 452.6 and 336.0 higher reward respectively.

**SPEDE-REG: Policy Optimization with SPEDE Representation Regularizer** In order to evaluate whether our assumption on linear MDP is valid in empirical settings and study whether such assumption can help improve the performance, we add our model dynamics representation objective as a regularizer in addition to the original SAC algorithm for learning the $Q$-function. Specifically, the algorithm SPEDE-REG consists of vanilla SAC objective with an additional loss putting constraints on the representation learned by the critic function, due to the intuition that the representation should satisfy the linear MDP dynamics. We compare its performance with the vanilla SAC algorithm to show the benefits of dynamic representation. Results in Table 2 show that adding such constraint significantly improve the performance of SAC: on hard tasks like Hopper-ET, S-Humanoid-ET and Humanoid-ET, SPEDE-REG improves the performance of SAC by 694.8, 1875.7 and 2000.4.

**Discussion of the Results** We observe that in the environments with relatively simple dynamics (top row of Table 2), the proposed SPEDE achieves the SoTA, even comparing with SAC. However, when the model dynamics of the environment become harder (bottom row of Table 2), the difference of the performance between the two approaches begin to enlarge. Interestingly, our SPEDE achieves strong results comparing to model-based approaches, while the joint learning SPEDE-REG outperforms model-free algorithm by a huge margin. The performance promotion of SPEDE indicates the importance on learning a good representation based on model dynamics and again shows the effectiveness of our approach in both settings.

In fact, the differences in the SoTA usage of SPEDE in easy environments and difficult environments also reveals the important direction for our future work. The current rigorous representation learning

---

[8]Our code is available at https://sites.google.com/view/spede.

Table 1: Performance of SPEDE on various MuJoCo control tasks. All the results are averaged across 4 random seeds and a window size of 10K. Results marked with * is directly adopted from MBBL (Wang et al., 2019). Our method achieves strong performance compared to pure empirical baselines (*e.g.*, PETS). We also compare SPEDE-REG which regularizes the critic using the model dynamics loss with several model-free RL method. SPEDE-REG significantly improves the performance of the SoTA method SAC.

| | Swimmer | Reacher | MountainCar | Pendulum | I-Pendulum |
|---|---|---|---|---|---|
| ME-TRPO* | 30.1±9.7 | -13.4±5.2 | -42.5±26.6 | **177.3±1.9** | -126.2±86.6 |
| PETS-RS* | 42.1±20.2 | -40.1±6.9 | -78.5±2.1 | 167.9±35.8 | -12.1±25.1 |
| PETS-CEM* | 22.1±25.2 | -12.3±5.2 | -57.9±3.6 | 167.4±53.0 | -20.5±28.9 |
| DeepSF | 25.5±13.5 | -16.8±3.6 | -17.0±23.4 | 168.6±5.1 | -0.2±0.3 |
| **SPEDE** | **42.6±4.2** | **-7.2±1.1** | **50.3±1.1** | 169.5±0.6 | **0.0±0.0** |
| PPO* | 38.0±1.5 | -17.2±0.9 | 27.1±13.1 | 163.4±8.0 | -40.8±21.0 |
| TRPO* | 37.9±2.0 | -10.1±0.6 | -37.2±16.4 | 166.7±7.3 | -27.6±15.8 |
| TD3* | 40.4±8.3 | -14.0±0.9 | -60.0±1.2 | 161.4±14.4 | -224.5±0.4 |
| SAC* | **41.2±4.6** | -6.4±0.5 | **52.6±0.6** | 168.2±9.5 | -0.2±0.1 |
| **SPEDE-REG** | 40.0±3.8 | **-5.8±0.6** | 40.0±3.8 | 168.5±4.3 | **0.0±0.1** |
| | Ant-ET | Hopper-ET | S-Humanoid-ET | Humanoid-ET | Walker-ET |
| ME-TRPO* | 42.6±21.1 | 4.9±4.0 | 76.1±8.8 | 72.9±8.9 | -9.5±4.6 |
| PETS-RS* | 130.0±148.1 | 205.8±36.5 | 320.9±182.2 | 106.9±106.9 | -0.8±3.2 |
| PETS-CEM* | 81.6±145.8 | 129.3±36.0 | 355.1±157.1 | 110.8±91.0 | -2.5±6.8 |
| DeepSF | 768.1±44.1 | 548.9±253.3 | 533.8±154.9 | 168.6±5.1 | 165.6±127.9 |
| **SPEDE** | **806.2±60.2** | **732.2±263.9** | **986.4±154.7** | **886.9±95.2** | **501.6±204.0** |
| PPO* | 80.1±17.3 | 758.0±62.0 | 454.3±36.7 | 451.4±39.1 | 306.1±17.2 |
| TRPO* | 116.8±47.3 | 237.4±33.5 | 281.3±10.9 | 289.8±5.2 | 229.5±27.1 |
| TD3* | 259.7±1.0 | 1057.1±29.5 | 1070.0±168.3 | 147.7±0.7 | **3299.7±1951.5** |
| SAC* | **2012.7±571.3** | 1815.5±655.1 | 834.6±313.1 | 1794.4±458.3 | 2216.4±678.7 |
| **SPEDE-REG** | **2073.1±119.7** | **2510.3±550.8** | **2710.3±277.5** | **3747.8±1078.1** | 2170.3±810.9 |

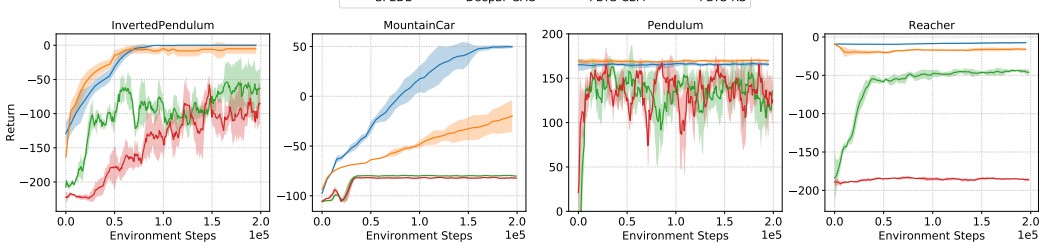

Figure 1: **Experiments on MuJoCo:** We show curves of the return versus the training steps for SPEDE and model-based RL baselines. Results show that in these tasks, our method enjoys better sample efficiency even compared to SoTA empirical model-based RL baselines.

methods, *e.g.*, Du et al. (2019); Misra et al. (2020); Agarwal et al. (2020a) and the proposed SPEDE, all rely on some model assumption. When the assumptions are satisfied, *e.g.*, Pendulum, Reacher, and others, our theoretically derived SPEDE variant works extremely well, even better than current SoTA. However, when the assumption is not fully satisfied, although the decoupled SPEDE achieves best performance among existing model-based RL and representation learning under fair comparison, the joint learned variant of SPEDE is more robust and promote the current SoTA with significant margin. An interesting question is whether we can rigorously justify the regularized SPEDE, which we leave as our future work.

**Performance Curves** To better understand how the sample complexity of our algorithm comparing to the prior model-based RL baselines, we plot the return versus environment steps in Figure 1. We see that comparing to prior model-based baselines, SPEDE enjoys great sample efficiency in these tasks. We want to emphasize that from MBBL (Wang et al., 2019), model-based methods already show significantly better sample efficiency compared to model-free methods (*e.g.*PPO/TRPO). We provide additional results in Appendix E.2.

## 6 CONCLUSION

In this paper, we introduce SPEDE, which, to the best of our knowledge, is the first provable and efficient representation learning algorithm for RL, by exploiting the benefits from noise. We provide thorough theoretical analysis for SPEDE and strong empirical results demonstrate the effectiveness of our algorithm. There are still several open problems, *e.g.*, how to rigorously justify the regularized SPEDE. We leave these problems as interesting future work.

**Ethics Statement** Our paper focus on one of the core questions in RL, *i.e.*, developing provable and practical algorithms for representation learning in online scenarios under general settings. Although there can be several potential application of our work in the future, currently we cannot foresee any positive/negative social impact of our work other than the original positive/negative social impact of RL. As we conduct our experiments on standard MuJoCo benchmark, our experiments should not suffer from any discrimination/bias/fairness concerns.

**Reproducibility Statement** Our code is directly adapted from the open-sourced code base from (Wang et al., 2019) and we have introduced our implementation details in Section 5 and Appendix E. We believe our results can be easily reproduced, and our code is available at https://sites.google.com/view/spede.

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

## A BACKGROUNDS ON REPRODUCING KERNEL HILBERT SPACE

We briefly introduce the basic concepts of the Reproducing Kernel Hilbert Space, which is helpful on understanding our paper. To start with, we first define the inner product (Stein & Shakarchi, 2009).

**Definition 6** (Inner Product). *A function $\langle \cdot, \cdot \rangle_{\mathcal{H}} : \mathcal{H} \times \mathcal{H} \to \mathbb{R}$ is said to be an inner product on $\mathcal{H}$ if it satisfies the following conditions:*

1. *Positive Definiteness: $\forall u \in \mathcal{H}$, $\langle u, u \rangle \geq 0$, and $\langle u, u \rangle = 0 \iff u = 0$.*

2. *Symmetry: $\forall u, v \in \mathcal{H}$, $\langle u, v \rangle \in \langle v, u \rangle$.*

3. *Bilinearity: $\forall \alpha, \beta \in \mathbb{R}, u, v, w \in \mathcal{H}$, $\langle \alpha u + \beta v, w \rangle = \alpha \langle u, w \rangle + \beta \langle v, w \rangle$.*

*Additionally, we can define a norm with the inner product: $\|u\| = \sqrt{\langle u, u \rangle}$.*

A Hilbert space is a space equipped with an inner product and satisfies an additional technical condition of completeness. The finite-dimension vector space with the canonical inner product is an example of the Hilbert space. We remark that $\mathcal{H}$ can also be a function space, for example, the space contains all square integrable functions (i.e. $\int_{\mathbb{R}} f(x)^2 \, dx < \infty$, generally denoted as $L_2$) is also a Hilbert space with inner product $\langle f, g \rangle = \int_{\mathbb{R}} f(x) g(x) \, dx$.

We then define the kernel, and introduce the notion of positive-definite kernel (Álvarez et al., 2012).

**Definition 7** ((Positive-Definite) Kernel). *A function $k : \mathcal{X} \to \mathcal{X} \to \mathbb{R}$ is said to be a kernel on non-empty set $\mathcal{X}$ if there exists a Hilbert space $\mathcal{H}$ and a feature map $\phi : \mathcal{X} \to \mathcal{H}$ such that $\forall x, x' \in \mathcal{X}$, we have*

$$k(x, x') = \langle \phi(x), \phi(x') \rangle_{\mathcal{H}}.$$

*Moreover, the kernel is said to be positive definite if $\forall n \geq 1$, $\forall \{a_i\}_{i \in [n]} \subset \mathbb{R}$ and mutually distinct set $\{x_i\}_{i \in [n]} \subset \mathcal{X}$, we have that*

$$\sum_{i \in [n]} \sum_{j \in [n]} a_i a_j k(x_i, x_j) > 0.$$

Some well-known kernels include:

- Linear Kernel: $k(x, y) = \langle x, y \rangle$, with the canonical feature map $\phi(x) = x$.
- Polynomial Kernel: $k(x, y) = (\langle x, y \rangle + c)^m$, where $m \in \mathbb{N}^+$ and $c \in \mathbb{R}^+$.
- Gaussian (a.k.a radial basis function, RBF) Kernel: $k(x, y) = \exp\left(\frac{\|x - y\|_2^2}{2\sigma^2}\right)$. It's known that such kernel is positive definite.

Now we can define the Reproducing Kernel Hilbert space (RKHS) (Aronszajn, 1950).

**Definition 8** (Reproducing Kernel Hilbert Space (RKHS)). *The Hilbert space $\mathcal{H}$ of $\mathbb{R}$-valued function defined on a non-empty set $\mathcal{X}$ is said to be a reproducing kernel Hilbert space (RKHS) is there is a kernel $k : \mathcal{X} \times \mathcal{X} \to \mathbb{R}$, such that*

1. *$\forall x \in \mathcal{X}$, $k(x, \cdot) \in \mathcal{H}$.*

2. *$\forall x \in \mathcal{X}, f \in \mathcal{H}$, $\langle f, k(x, \cdot) \rangle_{\mathcal{H}} = f(x)$ (a.k.a the reproducing property), which also implies that $\langle k(x, \cdot), k(y, \cdot) \rangle = k(x, y)$.*

*Here $k$ is called a reproducing kernel of $\mathcal{H}$.*

We provide an intuitive interpretation on the definition of RKHS when $\mathcal{H}$ is the space of linear function. Consider $\mathcal{X} = \mathbb{R}^d$ and $k(x, y) = \langle x, y \rangle$. With the definition of the kernel $k$, we can see that $k(x, \cdot) : \mathcal{X} \to \mathbb{R}$ is a linear function, and thus lies in $\mathcal{H}$. Meanwhile, $\forall f \in \mathcal{H}$, there exists $\theta_f$ such that $f(x) = \theta_f^\top x$. We define the inner product on $\mathcal{H}$ via $\langle f, g \rangle_{\mathcal{H}} = \langle \theta_f, \theta_g \rangle$, and thus $\langle f(, k(x, \cdot)) \rangle_{\mathcal{H}} = \theta^\top x = f(x)$, which demonstrates the reproducing property, and shows that the space of linear function on any finite-dimensional vector space is an RKHS with linear kernel as the corresponding reproducing kernel.

We state the following theorems without the proof.

**Theorem 9** (Moore-Aronszajn (Aronszajn, 1950)). *Every positive definite kernel $k$ is associated with a unique RKHS $\mathcal{H}$.*

Notice that, Moore-Aronszajn theorem guarantees that all of the positive kernel can be represented as the inner product in certain Hilbert space, hence we can have a linear representation of the Gaussian distribution induced by the reproducing property of Gaussian kernel, as we illustrated in the main text.

**Theorem 10** (Bochner (Rudin, 2017)). *A continuous, shift-invariant kernel (i.e. $k(x, y) = k(x - y)$) is positive definite if and only if $k(x - y)$ is the Fourier transform of a non-negative measure $\omega$, i.e.*

$$k(x - y) = \int_{\mathbb{R}^d} \exp(i\omega^\top (x - y)) \, d\mathbb{P}(\omega) = \int_{\mathbb{R}^d \times [0, 2\pi]} 2 \cos(\omega^\top x + b) \cos(\omega^\top y + b) \, d(\mathbb{P}(\omega) \times \mathbb{P}(b)),$$

*where $\mathbb{P}(b)$ is a uniform distribution on $[0, 2\pi]$.*

Bochner's theorem shows that any continuous positive definite shift-invariant kernel (e.g. Gaussian kernel, Laplacian kernel) can be represented as the inner product of random Fourier feature, which provides an additional way to provide a representation for certain distribution (see Rahimi & Recht, 2007; Dai et al., 2014).

## B   AN EQUIVALENT UPPER CONFIDENCE BOUND ALGORITHM

In this section, we provide a generic Upper Confidence Bound (UCB) algorithm with the OFU principle, and show the connections and differences between the UCB algorithm and the TS algorithm. The prototype for our UCB algorithm is illustrated in Algorithm 3.

---
**Algorithm 3** Upper Confidence Bound (UCB) Algorithm

---
**Require:** Number of Episodes $K$, Failure Probability $\delta \in (0, 1)$, Reward Function $r(s, a)$.
1: Initialize the history set $\mathcal{H}_0 = \emptyset$.
2: **for** episodes $k = 1, 2, \cdots$ **do**
3:     Compute $\pi_k$ via                                           ▷ Optimistic Planning.
$$(\pi_k, \tilde{f}_k) = \arg\max_{\pi \in \Pi, \tilde{f} \in \mathcal{F}_k} \tilde{V}_0^\pi(s_0).$$
    where $\mathcal{F}_k$ is defined in (9).
4:     **for** steps $h = 0, 1, \cdots, H - 1$ **do**                       ▷ Execute $\pi_k$.
5:         Execute $a_h^k \sim \pi_k^h(s_h^k)$.
6:         Observe $s_{h+1}$.
7:     **end for**
8:     Set $\mathcal{H}_k = \mathcal{H}_{k-1} \cup \{(s_h^k, a_h^k, s_{h+1}^k)\}_{h=0}^{H-1}$.             ▷ Update the History.
9: **end for**

---

Notice that, the only difference between UCB algorithm and TS algorithm is the mechanism of finding $f$ we use to plan for each episode (highlighted in blue). For UCB algorithm, we perform an optimistic planning, which finds the $\tilde{f}_k$ that potentially has the largest cumulative reward. However, such constrained optimization problem is NP-hard even for the simplest linear bandits (Dani et al., 2008). Instead, for TS algorithm, we only sample the $f_k$ from the posterior distribution, which gets rid of the complicated constraint optimization. We are interested in the UCB algorithm, as the worst case regret bound of the UCB algorithm can be directly translated to the expected regret bound of the TS algorithm without the need of explicit manipulation of the prior and the posterior(Russo & Van Roy, 2013; 2014; Osband & Van Roy, 2014).

**Confidence Set Construction**   Perhaps the most important part in OFU-style algorithm is the construction of confidence set $\mathcal{F}_k$. To enable sample-efficient learning, the confidence set should

1. contain $f^*$ with high probability, so that we can identify $f^*$ eventually;
2. shrink as fast as possible, so that we can identify $f^*$ efficiently.

In the tabular setting, $\mathcal{F}_k$ is constructed via the concentration of sub-Gaussian/sub-Gamma random variable (e.g. Azar et al., 2017; Jin et al., 2018; Zhang et al., 2021), and in the linear MDP setting, $\mathcal{F}_k$ is constructed via the concentration on the linear parameters. As we don't assume any

specific structures, we instead constructed $\mathcal{F}_k$ via the concentration on the $\ell_2$ error, following the idea of (Russo & Van Roy, 2013; Osband & Van Roy, 2014). Specifically, consider the least-square estimates defined by

$$\hat{f}_K = \arg\min_{f \in \mathcal{F}} L_{2,K}(f) := \sum_{k \in [K]} \sum_{h=0}^{H-1} \|f(s_h^k, a_h^k) - s_{h+1}^k\|_2^2. \tag{8}$$

As $s_{h+1}^k = f^*(s_h^k, a_h^k) + \epsilon_h^k$ where $\epsilon_h^k$ is the Gaussian noise added to the step $h$ at the $k$-th episode, we know $\hat{f}_K$ will not deviate from $f^*$ a lot. Meanwhile, as $K$ increases, the estimation $\hat{f}_K$ should become closer to $f^*$. Specifically, define the empirical 2-norm $\|\cdot\|_{2,E_t}$ as

$$\|g\|_{2,E_K}^2 := \sum_{k \in [K]} \sum_{h=0}^{H-1} \|g(s_h^k, a_h^k)\|_2^2.$$

We can construct the confidence set based on the following lemma:

**Lemma 11** (Confidence Set Construction (Russo & Van Roy, 2013; Osband & Van Roy, 2014)). *Define*

$$\mathcal{F}_K = \left\{ f \in \mathcal{F} : \|f - \hat{f}_K\|_{2,E_K} \le \sqrt{\beta_K^*(\mathcal{F}, \delta, \alpha)} \right\}, \tag{9}$$

*then*

$$\mathbb{P}_{f^*}\left( f^* \in \bigcap_{k=1}^{\infty} \mathcal{F}_k \right) \ge 1 - 2\delta, \tag{10}$$

*where*
$$\beta_K^*(\mathcal{F}, \delta, \alpha) = 8\sigma^2 \log(\mathcal{N}(\mathcal{F}, \alpha, \|\cdot\|_2)/\delta) + 2H\alpha(12C + \sqrt{8d\sigma^2 \log(4K^2H/\delta)}). \tag{11}$$

The proof can be found in Appendix C.1. Notice that, the empirical 2-norm $\|f - \hat{f}_K\|_{2,E_K}$ scales linearly with $K$, and $\beta_K^*(\mathcal{F}, \delta, \alpha)$ only scales as $\log K$, so the confidence set shrinks. Meanwhile, Equation 10 guarantees that $f^* \in \mathcal{F}_k, \forall k$ with high probability. Hence, it satisfies our requirement for the confidence set.

**Regret Upper Bound**  We have the following upper bound of the regret for the UCB algorithm:

**Theorem 12** (Regret Bound). *Assume Assumption 2 to 5 holds. We have that*
$$\text{Regret}(K) \le \tilde{O}(\sqrt{H^2 T \cdot \log \mathcal{N}(\mathcal{F}, T^{-1/2}, \|\cdot\|_2) \cdot \dim_E(\mathcal{F}, T^{-1/2})}).$$
*where $\tilde{O}$ represents the order up to logarithm factors.*

## C  TECHNICAL PROOF

### C.1  PROOF FOR LEMMA 11

*Proof.* We first show the following concentration on the $\ell_2$ error:

**Lemma 13** (Concentration of $\ell_2$ error (Russo & Van Roy, 2013; Osband & Van Roy, 2014; Wang et al., 2020)). $\forall \delta > 0, f : \mathcal{S} \times \mathcal{A} \to \mathbb{R}$, *we have*
$$\mathbb{P}_{f^*}\left( L_{2,K}(f) \ge L_{2,K}(f^*) + \frac{1}{2}\|f - f^*\|_{2,E_K}^2 - 4\sigma^2 \log(1/\delta), \quad \forall K \in \mathbb{N} \right) \ge 1 - \delta$$

*Proof.* Define the filtration $\mathcal{H}_{k,h} = \{(s_h^i, a_h^i)\}_{i \in [k-1], h=0, \cdots, H-1} \cup \{(s_i^k, a_i^k)\}_{h=0}^{h-1}$, and the random variable $Z_{k,h}$ adapted to the filtration $\mathcal{H}_{k,h}$ via:

$$\begin{aligned} Z_{k,h} &= \|f^*(s_h^k, a_h^k) - s_{h+1}^k\|_2^2 - \|f(s_h^k, a_h^k) - s_{h+1}^k\|_2^2 \\ &= \|f^*(s_h^k, a_h^k) - s_{h+1}^k\|_2^2 - \|f(s_h^k, a_h^k) - f^*(s_h^k, a_h^k) + f^*(s_h^k, a_h^k) - s_{h+1}^k\|_2^2 \\ &= -\|f(s_h^k, a_h^k) - f^*(s_h^k, a_h^k)\|_2^2 + 2\langle f(s_h^k, a_h^k) - f^*(s_h^k, a_h^k), \epsilon_h^k \rangle, \end{aligned}$$

where $\epsilon_h^k = s_{h+1}^k - f^*(s_h^k, a_h^k)$. Thus, $\mathbb{E}(Z_k^h|\mathcal{H}_{k,h}) = -\|f(s_h^k, a_h^k) - f^*(s_h^k, a_h^k)\|_2^2$, and $Z_k^h + \|f(s_h^k, a_h^k) - f^*(s_h^k, a_h^k)\|_2^2$ is a martingale w.r.t $\mathcal{H}_{k,h}$. Notice that we assume $\epsilon$ is an isotropic Gaussian noise with variance $\sigma^2$ on each of the dimension, thus the conditional moment generating function of $Z_k^h + \|f(s_h^k, a_h^k) - f^*(s_h^k, a_h^k)\|_2^2$ satisfies:

$$M_{k,h}(\lambda) = \log \mathbb{E}[\exp(\lambda(Z_k^h + \|f(s_h^k, a_h^k) - f^*(s_h^k, a_h^k)\|_2^2))|\mathcal{H}_{k,h}]$$

$$= \log \mathbb{E}[\exp(\langle 2\lambda f(s_h^k, a_h^k) - f^*(s_h^k, a_h^k), \epsilon_h^k\rangle)|\mathcal{H}_{k,h}]$$

$$\leq 2\sigma^2\lambda^2\|f(s_h^k) - f^*(s_h^k, a_h^k)\|_2^2.$$

Applying Lemma 4 in (Russo & Van Roy, 2013), we have that, $\forall x, \lambda \geq 0$,

$$\mathbb{P}_{f^*}\left(\sum_{k\in[K]}\sum_{h=0}^{H-1}\lambda Z_{k,h} \leq x - \lambda(1 - 2\lambda\sigma^2)\sum_{k\in[K]}\sum_{h=0}^{H-1}\|f(s_h^k, a_h^k) - f^*(s_h^k, a_h^k)\|_2^2, \quad \forall k \in \mathbb{N}\right) \leq 1 - \exp(-x).$$

Take $\lambda = \frac{1}{4\sigma^2}, x = \log 1/\delta$, and notice that $\sum_{k\in[K]}\sum_{h=0}^{H-1} Z_{k,h} = L_{2,K}(f^*) - L_{2,K}(f)$, we have the desired result. $\square$

We construct an $\alpha$-cover $\mathcal{F}_\alpha$ in $\mathcal{F}$ with respect to $\|\cdot\|_2$. With a standard union bound, we know that condition on $f^*$, with probability at least $1 - \delta$, we have that

$$L_{2,K}(f^\alpha) - L_{2,K}(f^*) \geq \frac{1}{2}\|f^\alpha - f^*\|_{2,E_K}^2 - 4\sigma^2 \log(|F^\alpha|/\delta), \quad \forall K \in \mathbb{N}, f^\alpha \in \mathcal{F}^\alpha.$$

Thus, we have that

$$L_{2,K}(f) - L_{2,K}(f^*) \geq \frac{1}{2}\|f - f^*\|_{2,E_K}^2 - 4\sigma^2 \log(|F^\alpha|/\delta)$$

$$+ \underbrace{\min_{f^\alpha\in\mathcal{F}^\alpha}\left\{\frac{1}{2}\|f^\alpha - f^*\|_{2,E_K}^2 - \frac{1}{2}\|f - f^*\|_{2,E_K}^2 + L_{2,K}(f) - L_{2,K}(f^\alpha)\right\}}_{\text{Discretization Error}}.$$

We then deal with the discretization error. Assume $\alpha \leq 2C$ (or otherwise we only have a trivial cover) and $\|f^\alpha(s, a) - f(s, a)\|_2 \leq \alpha$, we have that

$$\|f^\alpha(s, a) - f^*(s, a)\|_2^2 - \|f(s, a) - f^*(s, a)\|_2^2$$

$$= \|f^\alpha(s, a)\|_2^2 - \|f(s, a)\|_2^2 + 2\langle f^*(s, a), f(s, a) - f^\alpha(s, a)\rangle$$

$$\leq \max_{\|y\|_2\leq\alpha}\{\|f(s, a) + y\|_2^2 - \|f(s, a)\|_2^2\} + 2C\alpha$$

$$= \max_{\|y\|_2\leq\alpha}\{2\langle f(s, a), y\rangle + \|y\|_2^2\} + 2C\alpha$$

$$\leq 4C\alpha + \alpha^2 \leq 6C\alpha,$$

where the inequality is by Cauchy-Schwartz inequality and $\alpha \leq 2C$. Meanwhile,

$$\|s' - f(s, a)\|_2^2 - \|s' - f^\alpha(s, a)\|_2^2$$

$$= 2\langle s', f^\alpha(s, a) - f(s, a)\rangle + \|f(s, a)\|_2^2 - \|f^\alpha(s, a)\|_2^2$$

$$\leq 2\langle\epsilon, f^\alpha(s, a) - f(s, a)\rangle + 2\langle f^*(s, a), f^\alpha(s, a) - f(s, a)\rangle + 2C\alpha + \alpha^2$$

$$\leq 2\|\epsilon\|_2\alpha + 6C\alpha.$$

We now consider the concentration property of $\|\epsilon\|_2$. Here we simply follow (Jin et al., 2019) and notice that $\epsilon$ is $\sqrt{d}\sigma$-norm-sub-Gaussian, we have that

$$\mathbb{P}(\|\epsilon\|_2 > \sqrt{2d\sigma^2\log(2/\delta)}) \leq \delta.$$

By a union bound, we have that

$$\mathbb{P}(\exists k, \|\epsilon\|_2 > \sqrt{2d\sigma^2\log(4k^2H/\delta)}) \leq \frac{\delta}{2}\sum_{k=1}^\infty\sum_{h=0}^{H-1}\frac{1}{k^2H} \leq \delta.$$

Sum all these up, we can see with probability $1 - \delta$, $\forall K \in \mathbb{N}$, the discretization error is upper bounded by:

$$H\alpha(12C + \sqrt{8d\sigma^2\log(4K^2H/\delta)}).$$

As we consider the least square estimate $\hat{f}_K$, we have that $L_{2,K}(\hat{f}_K) - L_{2,K}(f^*) \leq 0$. Substitute back, we have the desired results. $\square$

## C.2 SIMULATION LEMMA

**Lemma 14** (Simulation Lemma (adapted from Lemma 3.9 in (Kakade et al., 2020))). *Given* $\hat{f}$, $\forall s \in \mathcal{S}$, *the value function* $\hat{V}^\pi$ *and* $V^\pi$ *corresponding to the model* $\hat{f}$ *and* $f^*$ *satisfies*

$$\hat{V}_0^\pi(s) - V_0^\pi(s) \leq H^{3/2}\sqrt{\mathbb{E}\left[\sum_{h=0}^{H-1}\min\left\{\frac{2\|f^*(s_h, a_h) - \hat{f}(s_h, a_h)\|_2^2}{\sigma^2}, 1\right\}\right]}.$$

*Proof.* We first show the following difference lemma:

**Lemma 15** (Difference Lemma). *Assume the trajectory $\{(s_h, a_h)\}_{h=0}^{H-1}$ is generated via policy $\pi$ and ground truth $f^*$, define*

$$V_h = \sum_{\tau=h}^{H-1} r(s_\tau, a_\tau)$$

*then $\forall \tau \in \{1, \cdots, H-1\}$, we have:*

$$\hat{V}_0^\pi(s_0) - V_0 = \mathbb{E}_{s_\tau' \sim \mathcal{N}(\hat{f}(s_{\tau-1}, a_{\tau-1}), \sigma^2 I)} \left[ \hat{V}_\tau^\pi(s_\tau') \right] - V_\tau$$

$$+ \sum_{h=1}^{\tau-1} \left[ \mathbb{E}_{s_h' \sim \mathcal{N}(f(s_{h-1}, a_{h-1}), \sigma^2 I)} \left[ \hat{V}_h^\pi(s_h') \right] - \hat{V}_h^\pi(s_h) \right].$$

*Proof.* When $\tau = 1$, we can obtain the result with $a_0 = \pi(s_0)$ and

$$\hat{V}_0^\pi(s_0) = r(s_0, \pi(s_0)) + \mathbb{E}_{s_1' \sim \mathcal{N}(f(s_0, a_0), \sigma^2 I)} \hat{V}_1^\pi(s_1').$$

We only need to show the case when $\tau = 2$, and the case when $\tau > 2$ can be derived via recursion. Notice that

$$\hat{V}_0^\pi(s_0) - V_0 = \mathbb{E}_{s_1' \sim \mathcal{N}(f(s_0, a_0), \sigma^2 I)} \left[ \hat{V}_1^\pi(s_1') \right] - V_1$$

$$= \hat{V}_1^\pi(s_1) - V_1 + \mathbb{E}_{s_1' \sim \mathcal{N}(f(s_0, a_0), \sigma^2 I)} \left[ \hat{V}_1^\pi(s_1') \right] - \hat{V}_1^\pi(s_1)$$

$$= \mathbb{E}_{s_2' \sim \mathcal{N}(f(s_1, a_1), \sigma^2 I)} \left[ \hat{V}_2^\pi(s_2') \right] - V_2 + \mathbb{E}_{s_1' \sim \mathcal{N}(f(s_0, a_0), \sigma^2 I)} \left[ \hat{V}_1^\pi(s_1') \right] - \hat{V}_1^\pi(s_1),$$

where the last equality is due to the fact that $a_1 = \pi(s_1)$. $\qquad\square$

We then follow the idea of "optional stopping" used in (Kakade et al., 2020) and show the following "optional stopping" simulation lemma.

**Lemma 16** ("Optional Stopping" Simulation Lemma). *Consider the stochastic process over the trajectories $\{(s_h, a_h)\}_{h=0}^{H-1}$ generated via policy $\pi$ and ground truth $f^*$, where the randomness is from the Gaussian noise in the dynamics. Define a stopping time $\tau$ w.r.t this stochastic process and a given model $\hat{f}$ via:*

$$\tau := \min\{h \geq 0 : \hat{V}_h^\pi(s_h) \leq V_h^\pi(s_h)\}.$$

*Furthermore, define a random variable:*

$$\tilde{V}_h^\pi(s_h) = \max\{\hat{V}_h^\pi(s_h), V_h^\pi(s_h)\},$$

*we have that*

$$\hat{V}_0^\pi(s_0) - V_0^\pi(s_0) \leq \mathbb{E}\left[ \sum_{h=0}^{H-1} \mathbf{1}_{h<\tau} \left( \mathbb{E}_{s_{h+1}' \sim \mathcal{N}(f^*(s_h, a_h), \sigma^2 I)} \tilde{V}_h^\pi(s_{h+1}') - \mathbb{E}_{s_{h+1}' \sim \mathcal{N}(\hat{f}(s_h, a_h), \sigma^2 I)} \tilde{V}_h^\pi(s_{h+1}') \right) \right],$$

*where the expectation is w.r.t the stochastic process over the trajectories.*

*Proof.* Define the filtration $\mathcal{F}_h := \{\epsilon_i\}_{i=0}^{h-1}$, where $\epsilon_i$ is the noise that add to the dynamics at step $i$. Define

$$M_h = \mathbb{E}[\hat{V}_0^\pi(s_0) - V_0 | \mathcal{F}_h],$$

which is a Doob martingale with respect to $\mathcal{F}_i$ (Grimmett & Stirzaker, 2020). As $\tau \leq H$, by Doob's optional stopping theorem, we have that

$$\mathbb{E}[\hat{V}_0^\pi(s_0) - V_0] = \mathbb{E}[M_\tau] = \mathbb{E}[\mathbb{E}[\hat{V}_0^\pi(s_0) - V_0 | \mathcal{F}_\tau]].$$

We then provide a bound for $M_\tau$. By Lemma 15, we have that

$$M_\tau = \mathbb{E}[\hat{V}_0^\pi(s_0) - V_0 | \mathcal{F}_\tau]$$

$$= \mathbb{E}_{s_\tau' \sim \mathcal{N}(\hat{f}(s_{\tau-1}, a_{\tau-1}), \sigma^2 I)} \left[ \hat{V}_\tau^\pi(s_\tau') \right] - V_\tau^\pi(s_\tau)$$

$$+ \mathbb{E}_{s_h' \sim \mathcal{N}(\hat{f}(s_{h-1}, a_{h-1}), \sigma^2 I)} \left[ \hat{V}_h^\pi(s_h') \right] - \sum_{h=1}^{\tau-1} \hat{V}_h^\pi(s_h)$$

$$= \sum_{h=1}^{\tau} \left( \mathbb{E}_{s_h' \sim \mathcal{N}(\hat{f}(s_h, a_h), \sigma^2)} \left[ \hat{V}_h^\pi(s_h') \right] - \tilde{V}_h^\pi(s_h) \right)$$

$$\leq \sum_{h=1}^{\tau} \left( \mathbb{E}_{s'_h \sim \mathcal{N}(\hat{f}(s_h,a_h),\sigma^2 I)} \left[ \tilde{V}_h^\pi(s_h) \right] - \tilde{V}_h^\pi(s_h) \right)$$

$$= \sum_{h=1}^{H} \mathbf{1}_{h \leq \tau} \left( \mathbb{E}_{s'_h \sim \mathcal{N}(\hat{f}(s_h,a_h),\sigma^2 I)} \left[ \tilde{V}_h^\pi(s_h) \right] - \tilde{V}_h^\pi(s_h) \right),$$

where the third inequality follows the definition of $\tau$ (and thus $V_\tau^\pi(s_\tau) = \tilde{V}_\tau^\pi(s_\tau)$ and $\hat{V}_h^\pi(s_h) = \tilde{V}_h^\pi(s_h)$ for $h < \tau$.)

The proof is then concluded via the following observation:

$$\mathbb{E} \left[ \mathbf{1}_{h \leq \tau} \left( \mathbb{E}_{s'_h \sim \mathcal{N}(\hat{f}(s_h,a_h),\sigma^2 I)} \left[ \tilde{V}_h^\pi(s_h) \right] - \tilde{V}_h^\pi(s_h) \right) \right]$$

$$= \mathbb{E} \left[ \mathbb{E} \left[ \mathbf{1}_{h \leq \tau} \left( \mathbb{E}_{s'_h \sim \mathcal{N}(\hat{f}(s_h,a_h),\sigma^2 I)} \left[ \tilde{V}_h^\pi(s_h) \right] - \tilde{V}_h^\pi(s_h) \right) \Big| \mathcal{F}_{h-1} \right] \right]$$

$$= \mathbb{E} \left[ \mathbb{E} \left[ \mathbf{1}_{h-1 < \tau} \left( \mathbb{E}_{s'_h \sim \mathcal{N}(\hat{f}(s_h,a_h),\sigma^2 I)} \left[ \tilde{V}_h^\pi(s_h) \right] - \tilde{V}_h^\pi(s_h) \right) \Big| \mathcal{F}_{h-1} \right] \right]$$

$$= \mathbb{E} \left[ \mathbf{1}_{h-1 < \tau} \mathbb{E} \left[ \left( \mathbb{E}_{s'_h \sim \mathcal{N}(\hat{f}(s_h,a_h),\sigma^2 I)} \left[ \tilde{V}_h^\pi(s_h) \right] - \tilde{V}_h^\pi(s_h) \right) \Big| \mathcal{F}_{h-1} \right] \right]$$

$$= \mathbb{E} \left[ \mathbf{1}_{h-1 < \tau} \left( \mathbb{E}_{s'_h \sim \mathcal{N}(\hat{f}(s_h,a_h),\sigma^2)} \left[ \tilde{V}_h^\pi(s_h) \right] - \mathbb{E}_{s'_h \sim \mathcal{N}(f^*(s_h,a_h),\sigma^2)} \left[ \tilde{V}_h^\pi(s_h) \right] \right) \right],$$

where the third equality is due to the fact that $\mathbf{1}_{h-1 < \tau}$ is measurable under $\mathcal{F}_{h-1}$. $\qquad\square$

Before we finally provide the proof of Lemma 14, we state the following lemma that bound the expectation under two isotropic Gaussian distribution with different mean:

**Lemma 17** (Difference of Expectation under Different Mean Isotropic Gaussian). $\forall$ *(approximately measurable) positive function g, we have that*

$$\mathbb{E}_{z \sim \mathcal{N}(\mu_1,\sigma^2 I)}[g(z)] - \mathbb{E}_{z \sim \mathcal{N}(\mu_2,\sigma^2 I)}[g(z)] \leq \min \left\{ \frac{\sqrt{2}\|\mu_1 - \mu_2\|}{\sigma}, 1 \right\} \sqrt{\mathbb{E}_{z \sim \mathcal{N}(\mu_1,\sigma^2 I)}[g(z)^2]}$$

*Proof.*

$$\mathbb{E}_{z \sim \mathcal{N}(\mu_1,\sigma^2 I)}[g(z)] - \mathbb{E}_{z \sim \mathcal{N}(\mu_2,\sigma^2 I)}[g(z)]$$

$$= \mathbb{E}_{z \sim \mathcal{N}(\mu_1,\sigma^2 I)} \left[ g(z) \left( 1 - \exp \left( \frac{2(\mu_1 - \mu_2)^\top z + \|\mu_2\|^2 - \|\mu_1\|^2}{2\sigma^2} \right) \right) \right]$$

$$\leq \sqrt{\mathbb{E}_{z \sim \mathcal{N}(\mu_1,\sigma^2 I)}[g(z)^2]} \sqrt{\mathbb{E}_{z \sim \mathcal{N}(\mu_1,\sigma^2 I)} \left( 1 - \exp \left( \frac{2(\mu_2 - \mu_1)^\top z - \|\mu_2\|^2 + \|\mu_1\|^2}{2\sigma^2} \right) \right)^2}$$

We then calculate

$$\mathbb{E}_{z \sim \mathcal{N}(\mu_1,\sigma^2 I)} \left( 1 - \exp \left( \frac{2(\mu_2 - \mu_1)^\top z - \|\mu_2\|^2 + \|\mu_1\|^2}{2\sigma^2} \right) \right)^2$$

$$= 1 - \frac{2}{\sqrt{2\pi}\sigma^{d/2}} \int \exp \left( \frac{-\|z - \mu_1\|_2^2 + 2(\mu_2 - \mu_1)^\top z - \|\mu_2\|^2 + \|\mu_1\|^2}{2\sigma^2} \right) dz$$

$$+ \frac{1}{\sqrt{2\pi}\sigma^{d/2}} \int \exp \left( \frac{-\|z - \mu_1\|_2^2 + 4(\mu_2 - \mu_1)^\top z - 2\|\mu_2\|^2 + 2\|\mu_1\|^2}{2\sigma^2} \right) dz$$

$$= -1 + \frac{1}{\sqrt{2\pi}\sigma^{d/2}} \int \exp \left( \frac{-\|z - (2\mu_2 - \mu_1)\|_2^2 + 2\|\mu_2 - \mu_1\|_2^2}{2\sigma^2} \right) dz$$

$$= -1 + \exp \left( \frac{\|\mu_2 - \mu_1\|_2^2}{\sigma^2} \right).$$

Also notice that, as $g$ is positive, a simple bound is that

$$\mathbb{E}_{z \sim \mathcal{N}(\mu_1,\sigma^2 I)}[g(z)] - \mathbb{E}_{z \sim \mathcal{N}(\mu_2,\sigma^2 I)}[g(z)] \leq \mathbb{E}_{z \sim \mathcal{N}(\mu_1,\sigma^2 I)}[g(z)] \leq \sqrt{\mathbb{E}_{z \sim \mathcal{N}(\mu_1,\sigma^2 I)}[g(z)^2]}.$$

Thus,

$$\mathbb{E}_{z \sim \mathcal{N}(\mu_1,\sigma^2 I)}[g(z)] - \mathbb{E}_{z \sim \mathcal{N}(\mu_2,\sigma^2 I)}[g(z)] \leq \sqrt{\mathbb{E}_{z \sim \mathcal{N}(\mu_1,\sigma^2 I)}[g(z)^2]} \sqrt{\min \left\{ \exp \left( \frac{\|\mu_2 - \mu_1\|_2^2}{\sigma^2} \right) - 1, 1 \right\}}.$$

Notice that, if $\|\mu_2 - \mu_1\| \geq \sigma$, then $\exp\left(\frac{\|\mu_2-\mu_1\|_2^2}{\sigma^2}\right) - 1 \geq 1$. Meanwhile, when $x \in [0,1]$, $\exp(x) \leq 1 + 2x$. Thus,

$$\sqrt{\min\left\{\exp\left(\frac{\|\mu_2 - \mu_1\|_2^2}{\sigma^2}\right) - 1, 1\right\}} \leq \sqrt{\min\left\{1 + \frac{2\|\mu_2 - \mu_1\|_2^2}{\sigma^2} - 1, 1\right\}} = \min\left\{\frac{2\|\mu_2 - \mu_1\|^2}{\sigma^2}, 1\right\},$$

which finishes the proof. $\square$

With Lemma 16, we have that
$$\hat{V}_0^\pi(s_0) - V_0^\pi(s_0)$$

$$\leq \mathbb{E}\left[\mathbf{1}_{h-1<\tau}\left(\mathbb{E}_{s_h'\sim\mathcal{N}(\hat{f}(s_h,a_h),\sigma^2)}\left[\tilde{V}_h^\pi(s_h)\right] - \mathbb{E}_{s_h'\sim\mathcal{N}(f^*(s_h,a_h),\sigma^2)}\left[\tilde{V}_h^\pi(s_h)\right]\right)\right]$$

$$\leq \sum_{h=0}^{H-1}\mathbb{E}\left[\sqrt{\mathbb{E}_{s_{h+1}'\sim\mathcal{N}(\hat{f}(s_h,a_h),\sigma^2)}\left[\tilde{V}_h^\pi(s_{h+1}')^2\right]}\min\left\{\frac{\sqrt{2}\|f^*(s_h,a_h) - \hat{f}(s_h,a_h)_2\|}{\sigma}, 1\right\}\right]$$

$$\leq \sum_{h=0}^{H-1}\sqrt{\mathbb{E}\left[\mathbb{E}_{s_{h+1}'\sim\mathcal{N}(\hat{f}(s_h,a_h),\sigma^2)}\left[\tilde{V}_h^\pi(s_{h+1}')^2\right]\right]}\sqrt{\mathbb{E}\left[\min\left\{\frac{2\|f^*(s_h,a_h) - \hat{f}(s_h,a_h)_2^2\|}{\sigma^2}, 1\right\}\right]}$$

$$\leq \sqrt{\mathbb{E}\left[\sum_{h=0}^{H-1}\mathbb{E}_{s_{h+1}'\sim P(\cdot|f^*(s_h,a_h))}\left[\tilde{V}_h^\pi(s_{h+1}')^2\right]\right]}\sqrt{\mathbb{E}\left[\sum_{h=0}^{H-1}\min\left\{\frac{2\|f^*(s_h,a_h) - \hat{f}(s_h,a_h)_2^2\|}{\sigma^2}, 1\right\}\right]}$$

$$\leq H^{3/2}\sqrt{\mathbb{E}\left[\sum_{h=0}^{H-1}\min\left\{\frac{2\|f^*(s_h,a_h) - \hat{f}(s_h,a_h)_2^2\|}{\sigma^2}, 1\right\}\right]}$$

where the second inequality is due to Lemma 17, and the last inequality is due to the fact that $\tilde{V}_h^\pi(s_{h+1}') \leq H, \forall h$. $\square$

## C.3 SUM OF WIDTH SQUARE

**Lemma 18** (Bound on the Sum of Width Square). *Define*
$$w_\mathcal{F}(s,a) := \sup_{\bar{f},\underline{f}\in\mathcal{F}}\|\bar{f}(s,a) - \underline{f}(s,a)\|_2.$$
*If $\{\beta_k^*\}_{k\in[K]}$ is a non-decreasing sequence, and $\|f\|_2 < C, \forall f \in \mathcal{F}$, then:*
$$\sum_{k\in[K]}\sum_{h=0}^{H-1}w_{\mathcal{F}_t}^2(s_h^k, a_h^k) \leq 1 + 4C^2 H\dim_E\left(\mathcal{F}, T^{-1/2}\right) + 4\beta_K\dim_E\left(\mathcal{F}, T^{-1/2}\right)(1 + \log T)$$

*Proof.* We first show the following lemma, which will be helpful in our proof.

**Lemma 19** (Lemma 1 in (Osband & Van Roy, 2014)). *If $\{\beta_k\}_{k\in[K]}$ is a non-decreasing sequence, we have*
$$\sum_{k\in[K]}\sum_{h=0}^{H-1}\mathbf{1}_{w_{\mathcal{F}_k}(s_h^k, a_h^k)>\epsilon} \leq \left(\frac{4\beta_K}{\epsilon^2} + H\right)\dim_E(\mathcal{F}, \epsilon).$$

*Proof.* We first consider when $w_{\mathcal{F}_k}(s_h^k, a_h^k) > \epsilon$ and is $\epsilon$-dependent on $n$ disjoint sub-sequences of $\{(s_h^i, a_h^i)\}_{i\in[k-1]}$. By the definition of $\epsilon$-dependent, we know $\|\bar{f} - \underline{f}\|_{2,E_k} > n\epsilon^2$. On the other hand, by triangle inequality, we know $\|\bar{f} - \underline{f}\|_{2,E_k} \leq 2\sqrt{\beta_k} \leq 2\sqrt{\beta_K}$, thus $n < \frac{4\beta_K}{\epsilon^2}$. Hence we know when $w_{\mathcal{F}_k}(s_h^k, a_h^k) > \epsilon$, then $(s_h, a_h)$ is at most $\epsilon$-dependent on $\frac{4\beta_K}{\epsilon^2}$ disjoint sub-sequences of $\{(s_h^i, a_h^i)\}^{i\in[k-1]}$.

We then show that, for any sequence $\{(s_i, a_i)\}_{i\in[N]}$, there is some element $(s_j, a_j)$ that is $\epsilon$-dependent on at least $\frac{n}{\dim_E(\mathcal{F},\epsilon)} - H$ disjoint sub-sequences of $\{(s_i, a_i)\}_{i\in[j-1]}$. Let $n$ satisfies that $n\dim_E(\mathcal{F}, \epsilon) + 1 \leq N \leq (n+1)\dim_E(\mathcal{F}, \epsilon)$, and we will construct $n$ disjoint sub-sequences

$\{B_i\}_{i\in[n]}$. We first let $B_i = \{(s_i, a_i)\}, \forall i \in [n]$. If $(s_{k+1}, a_{k+1})$ is $\epsilon$-dependent on each $B_i, i \in [n]$, we have the desired results. Otherwise, we append $(s_{k+1}, a_{k+1})$ to the sub-sequence that it is $\epsilon$-independent with. Repeat this process until some $j > n + 1$ is $\epsilon$-dependent on each sub-sequence or we have reached $N$. In the latter case we have $\sum_{i\in[n]} |B_i| \geq n\dim_E(\mathcal{F}, \epsilon)$ (here we can add at most $H - 1$ data to avoid the case we need a new episode of data), and since each element of a sub-sequence is $\epsilon$-independent with its predecessors, $|B_i| \leq \dim_E(\mathcal{F}, \epsilon), \forall i$ by the definition of eluder dimension. Thus $|B_i| = \dim_E(\mathcal{F}, \epsilon), \forall i$. And in this case, $(s_N, a_N)$ must be $\epsilon$-dependent on each sub-sequence by the definition of eluder dimension. Notice that, as our data is collected in an episodic pattern, there are at most $H - 1$ sub-sequences that contains "imaginary" final episode data introduced to the construction. In this case, we know that there are at least $\frac{n}{\dim_E(\mathcal{F},\epsilon)} - H$ disjoint sub-sequences that $(s_N, a_N)$ is $\epsilon$-dependent, which finishes our claim.

We finally consider the sub-sequence $B = \{(s_h^k, a_h^k)\}$ with $w_{\mathcal{F}_k}(s_h^k, a_h^k) > \epsilon$. We know that each element in $B$ is $\epsilon$-dependent on at most $\frac{4\beta_K}{\epsilon^2}$ disjoint sub-sequence of $B$, but at least $\epsilon$-dependent on $\frac{|B|}{\dim_E(\mathcal{F},\epsilon)} - H$ sub-sequence of $B$. Thus we know $|B| \leq \left(\frac{4\beta_K}{\epsilon^2} + H\right)\dim_E(\mathcal{F}, \epsilon)$, which concludes the proof. $\qquad\square$

For notation simplicity, we define $w_{t,h} := w_{\mathcal{F}_t}(s_h^t, a_h^t)$. We first reorder the sequence $\{w_{t,h}\}_{k\in[K], 0\leq h\leq H-1} \to \{w_i\}_{i\in[KH]}$, such that $w_1 \geq \cdots w_{TH}$. Then we have

$$\sum_{k\in[K]}\sum_{h=0}^{H-1} w_{\mathcal{F}_t}^2(s_h^k, a_h^k) = \sum_{i\in[KH]} w_i^2 \leq \sum_{i\in[KH]} w_i^2 \mathbf{1}_{w_i < T^{-1/2}} + \sum_{i\in[KH]} w_i^2 \mathbf{1}_{w_i \geq T^{-1/2}} \leq 1 + \sum_{i\in[KH]} w_i^2 \mathbf{1}_{w_i \geq T^{-1/2}}.$$

As we order the sequence, $w_j \geq \epsilon$ means

$$\sum_{k\in[K]}\sum_{h=0}^{H-1} \mathbf{1}_{w_{\mathcal{F}_t}(s_h^k, a_h^k) > \epsilon} \geq j.$$

Hence we know

$$\epsilon \leq \sqrt{\frac{4\beta_K}{\frac{j}{\dim_E(\mathcal{F},\epsilon)} - H}} = \sqrt{\frac{4\beta_K \dim_E(\mathcal{F}, \epsilon)}{j - H\dim_E(\mathcal{F}, \epsilon)}},$$

which means if $w_i \geq T^{-1/2}$, then $w_i < \min\left\{2C, \sqrt{\frac{4\beta_K \dim_E(\mathcal{F}, T^{-1/2})}{k - H\dim_E(\mathcal{F}, T^{-1/2})}}\right\}$. Hence,

$$\sum_{i\in[KH]} w_i^2 \mathbf{1}_{w_i \geq T^{-1/2}} \leq 4C^2 H\dim_E\left(\mathcal{F}, T^{-1/2}\right) + \sum_{j=H\dim_E(\mathcal{F}, T^{-1/2})+1}^{T} \frac{4\beta_K \dim_E(\mathcal{F}, T^{-1/2})}{j - H\dim_E(\mathcal{F}, T^{-1/2})}$$

$$\leq 4C^2 H\dim_E\left(\mathcal{F}, T^{-1/2}\right) + 4\beta_K \dim_E\left(\mathcal{F}, T^{-1/2}\right)(1 + \log T),$$

which finishes the proof. $\qquad\square$

## C.4 PROOF FOR THEOREM 5 AND THEOREM 12

*Proof.* Define $\mathcal{E}_k = \mathbb{P}_{f^*}(f^* \in \mathcal{F}_k)$. When constructing the confidence set, take $\alpha = T^{-1/2}$ and $\delta = 0.25$ in Lemma 11, which leads to

$$\beta_k^* := 8\sigma^2 \log(4\mathcal{N}(\mathcal{F}, T^{-1/2}, \|\cdot\|_2)) + HT^{-1/2}(12C + \sqrt{8d\sigma^2 \log(16k^2 H)}).$$

With our confidence set construction, we know that $\sum_{k\in[K]} P(\bar{\mathcal{E}}_k) \leq 0.5$. Notice that

$$\text{Regret}(K) = \sum_{k\in[K]} \left[V_0^*(s_0^k) - V_0^{\pi_k}(s_0^k)\right]$$

$$\leq \mathbb{E}\left[\sum_{k\in[K]} \mathbb{E}\left[\mathbb{P}(\mathcal{E}_k)[V^*(s_0^k) - V_0^{\pi_k}(s_0^k)]\right]\right] + H\sum_{k\in[K]} \mathbb{P}(\bar{\mathcal{E}}_k)$$

$$\leq \mathbb{E}\left[\sum_{k\in[K]} \mathbb{E}\left[\tilde{V}_{0,k}^{\pi_k}(s_0^k) - V_0^{\pi_k}(s_0^k)\right]\right] + 0.5H$$

$$\leq H^{3/2} \sum_{k \in K} \sqrt{\mathbb{E}\left[\sum_{h=0}^{H-1} \min\left\{\frac{2\|\tilde{f}_k(s_h^k, a_h^k) - f^*(s_h^k, a_h^k)\|_2^2}{\sigma^2}, 1\right\}\right]} + 0.5H$$

$$\leq \sqrt{H^2 T \mathbb{E}\left[\sum_{k \in [K]} \sum_{h=0}^{H-1} \min\left\{\frac{2\|\tilde{f}_k(s_h^k, a_h^k) - \hat{f}^*(s_h^k, a_h^k)\|_2^2}{\sigma^2}, 1\right\}\right]} + 0.5H$$

$$\leq \sqrt{\frac{2H^2 T}{\sigma^2}\left(1 + 4C^2 H \dim_E\left(\mathcal{F}, T^{-1/2}\right) + 4\beta_K^* \dim_E\left(\mathcal{F}, T^{-1/2}\right)(1 + \log T)\right)} + 0.5H,$$

where the first equality is due to the fact that the total reward for each episode is bounded in $[0, H]$, the second inequality is due to the optimism and our confidence set construction, the third inequality is due to Lemma 14, the fourth inequality is due to Cauchy-Schwartz inequality and the final inequality is due to Lemma 18 , which concludes the proof of Theorem 12. Following the idea of (Russo & Van Roy, 2013; 2014; Osband & Van Roy, 2014), we can translate the worst-case regret bound for UCB algorithm into the expected regret bound for TS algorithm, that conclude the proof of Theorem 5. □

**Remark** It can be undesirable that our regret bound scale with $\sigma^{-1}$, which means our algorithm can perform pretty bad when the noise level is extremely low. It is also more or less counter-intuitive. We want to remark that, such phenomenon is only an artifact introduced by our proof strategy. The simulation lemma (Lemma 14) works well when $f(s, a) - \tilde{f}(s, a)$ is small. However, we need to tolerate some bad episodes to collect sufficient samples, that can eventually make the error small. Fortunately, the regret of such bad episode is at most $H$. Hence, we can use the following strategy to get rid of the dependency on $\sigma^{-1}$.

**Definition 20** (Bad and Good Episodes)**.** *Define episode $k$ as a bad episode, if $\exists h \in \{0, 1, \cdots, H - 1\}$, such that $w_{k,h} := w_{\mathcal{F}_k}(s_h^k, a_h^k)$ is the largest $H \dim_E(\mathcal{F}, \sigma^2 T^{1/2})$ elements in the set $\{w_{k,h}\}_{k \in [K], 0 \leq h \leq H-1}$. Define episode $k$ as a good episode, if it is not a bad episode.*

By the definition, we know there are at most $H \dim_E(\mathcal{F}, \sigma^2 T^{-1/2})$ bad episodes. We then show the following lemma, that can be directly generalized from Lemma 18, by setting $\epsilon = \sigma^2 T^{-1/2}$ and remove the terms from bad episodes.

**Lemma 21.** *If $\{\beta_k^*\}_{k \in [K]}$ is a non-decreasing sequence, and $\|f\|_2 < C, \forall f \in \mathcal{F}$, then:*

$$\sum_{k \in [K], k \text{ is good}} \sum_{h=0}^{H-1} w_{\mathcal{F}_t}^2(s_h^k, a_h^k) \leq \sigma^2 + 4\beta_K \dim_E\left(\mathcal{F}, \sigma^2 T^{-1/2}\right)(1 + \log T)$$

Eventually, we can obtain the following regret bound, by setting the regret of bad episodes as $H$, and bounding the regret of good episodes with Lemma 21.

**Theorem 22** (Improved Regret Bound)**.** *Assume Assumption 2 to 5 holds. Take $\alpha = \sigma^2 T^{-1/2}$ and $\delta = 0.25$ in Lemma 11, which leads to*

$$\beta_k^* := 8\sigma^2 \log(4\mathcal{N}(\mathcal{F}, \sigma^2 T^{-1/2}, \|\cdot\|_2)) + H\sigma^2 T^{-1/2}(12C + \sqrt{8d\sigma^2 \log(16k^2 H)}).$$

*We have that*

$$\text{Regret}(K) \leq \sqrt{H^2 T\left(\frac{8\beta_K}{\sigma^2} + 1\right)\dim_E(\mathcal{F}, \sigma^2 T^{-1/2})(1 + \log T)} + 0.5H + H^2 \dim_E(\mathcal{F}, \sigma^2 T^{-1/2})$$

We would like to remark, that the definition of bad and good episodes is only used for the proof. We don't need to make any modification on the algorithm. Notice that, as $\beta_k^* \propto \sigma^2$, our upper bound in 22 can only scale with $\sigma^{-1}$ through the logarithm covering number $\log(4\mathcal{N}(\mathcal{F}, \sigma^2 T^{-1/2})$ and eluder dimension $\dim_E(\mathcal{F}, \sigma^2 T^{-1/2})$. When $\mathcal{F}$ is a linear function class, both term should scale with $\text{polylog}(\sigma)$, that matches the result from (Kakade et al., 2020).

## D  BOUNDS ON THE COMPLEXITY TERM UNDER LINEAR REALIZABILITY

We provide the upper bound on the covering number and the eluder dimension of $\mathcal{F}$ when $\mathcal{F} := \{\theta^\top \varphi : \theta \in \mathbb{R}^{d_\varphi \times d}, \|\theta\|_2 \leq W\}$ where $\varphi : \mathcal{S} \times \mathcal{A} \to \mathbb{R}^{d_\varphi}$ is some known feature map. We first make the following standard assumption:

**Assumption 6** (Bounded Feature).
$$\|\varphi(s,a)\|_2 \le B, \forall (s,a) \in \mathcal{S} \times \mathcal{A}.$$

### D.1 COVERING NUMBER

**Theorem 23** (Covering Number Bound). *We have that*
$$\mathcal{N}(\mathcal{F}, \epsilon, \|\cdot\|_2) \le \left(1 + \frac{2BW}{\epsilon}\right)^{d_\varphi}.$$

*Proof.* Notice that, by Cauchy-Schwartz inequality, we have that
$$\max_{(s,a) \in \mathcal{S} \times \mathcal{A}} \|\varepsilon_i^\top \varphi(s,a)\|_2 \le B\|\varepsilon_i\|_2, \quad \forall \varepsilon_i \in \mathbb{R}^{d_\varphi}.$$
Thus, denote $\varepsilon = [\varepsilon_i]_{i \in [d]}$, we have that
$$\max_{(s,a) \in \mathcal{S} \times \mathcal{A}} \|\varepsilon^\top \varphi(s,a)\|_2^2 = \max_{(s,a) \in \mathcal{S} \times \mathcal{A}} \sum_{i \in [d]} \|\varepsilon_i^\top \varphi(s,a)\|_2^2 \le B^2 \sum_{i \in [d]} \|\varepsilon_i\|_2^2 = B^2 \|\varepsilon\|_2^2.$$
Hence, to find an $\epsilon$-cover for $\mathcal{F}$, we just need to find an $\epsilon/B$-cover of $\{\theta : \theta \in \mathbb{R}^{d_\varphi \times d}, \|\theta\|_2 \le W\}$. By standard argument on the covering number of Euclidean space (e.g. Lemma 5.7 in (Wainwright, 2019)), we can conclude the desired result. $\square$

### D.2 ELUDER DIMENSION

**Theorem 24** (Eluder Dimension Bound). *We have that*
$$\dim_E(\mathcal{F}, \epsilon) \le \frac{3d_\varphi e}{e-1} \log\left(3 + \frac{12W^2 B^2}{\epsilon^2}\right) + 1.$$

*Proof.* Our proof follows the idea in (Russo & Van Roy, 2013). Define
$$w_k := \sup\left\{(\theta_1 - \theta_2)^\top \varphi(s,a) : \sqrt{\sum_{i \in [k-1]} ((\theta_1 - \theta_2)^\top \varphi_i(s_i, a_i))^2} \le \epsilon', \theta_1, \theta_2 \in \mathbb{R}^{d_\varphi \times d}, \|\theta_1\| \le W, \|\theta_2\| \le W\right\}.$$
For notation simplicity, define $\varphi_k := \varphi(s_i, a_i), \theta := \theta_1 - \theta_2$, and $\Phi_k := \sum_{i \in [k-1]} \varphi_i \varphi_i^\top$. Obviously, we have that $\|\theta\| \le 2W$. Moreover, by straightforward calculation, we know
$$\sum_{i \in [k-1]} \left((\theta_1 - \theta_2)^\top \varphi_i(s_i, a_i)\right)^2 = \text{Trace}(\theta^\top \varphi_k \theta).$$
Define $V_k := \Phi_k + \frac{(\epsilon')^2}{4W^2} I$, we start from considering the problem
$$\max_\theta \text{Trace}(\theta^\top \varphi_k \varphi_k^\top \theta), \quad \text{subject to} \quad \text{Trace}(\theta^\top V_k \theta) \le 2\epsilon^2.$$
The Lagrangian can be formed as
$$\mathcal{L}(\theta, \gamma) = -\text{Trace}(\theta^\top \varphi_k \varphi_k^\top \theta) + \lambda(\text{Trace}(\theta^\top V_k \theta) - 2\epsilon^2), \quad \lambda \ge 0.$$
The optimality condition of $\theta$ is
$$(\lambda V_k - \varphi_k \varphi_k^\top)\theta = 0.$$
As $V_k$ is of full rank, $\lambda V_k - \varphi_k \varphi_k^\top$ has rank at least $d_\varphi - 1$ (as $\varphi_k \varphi_k^\top$ is of rank 1). So the equation
$$(\lambda V_k - \varphi_k \varphi_k^\top)\theta_i = 0, \quad \theta_i \in \mathbb{R}^{d_\varphi}$$
only has one non-zero solution. Substitute back, we know that (define $\|x\|_A := \sqrt{x^\top A x}$):
$$\sup\{\text{Trace}(\theta^\top \varphi_k \varphi_k^\top \theta) : \text{Trace}(\theta^\top V_k \theta) \le \epsilon^2\} = \sqrt{2}\epsilon' \|\varphi_k\|_{V_k^{-1}}.$$
With the conclusion above, we have that
$$w_k \le \sup\{\theta^\top \varphi_k : \text{Trace}(\theta^\top \Phi_k \theta) \le \epsilon^2, \|\theta\| \le 2W\} \le \sup\{\theta^\top \varphi_k : \text{Trace}(\theta^\top V_k \theta) \le 2\epsilon^2\} = \sqrt{2}\epsilon' \|\varphi_k\|_{V_k^{-1}}.$$
Hence, if $w_k \ge \epsilon'$, then $\varphi_k V_k^{-1} \varphi_k \ge 0.5$. Moreover, with Matrix Determinant Lemma, if $w_i \ge \epsilon'$, $\forall i < k$, we have
$$\det(V_k) = \det(V_{k-1})(1 + \varphi_k^\top V_k^{-1} \varphi_k) \ge \det(V_{k-1})\left(\frac{3}{2}\right) \ge \cdots \ge \det\left(\frac{(\epsilon')^2}{4W^2}I\right)\left(\frac{3}{2}\right)^{k-1} = \frac{(\epsilon')^{2d}}{4W^{2d}}\left(\frac{3}{2}\right)^{k-1}.$$
Meanwhile,
$$\det(V_k) \le \left(\frac{\text{Trace}(V_k)}{d}\right)^d \le \left(\frac{B^2(k-1)}{d} + \frac{(\epsilon')^2}{4W^2}\right)^d.$$

Hence, we know

$$\left(\frac{3}{2}\right)^{(k-1)/d} \le \frac{4W^2B^2}{(\epsilon')^2} \cdot \frac{k-1}{d} + 1.$$

Now we only need to find the largest $k$ that can make this inequality hold. For notation simplicity, define $\alpha := \frac{4W^2B^2}{(\epsilon')^2}$, $n = \frac{k-1}{d}$. As $\log(1+x) \ge \frac{x}{1+x}$ and $\log x \le x/e$, we have

$$\frac{n}{3} \le n \log 3/2 \le \log(\alpha+1) + \log n \le \log(\alpha+1) + \log 3 + \log(n/3) \le \log(\alpha+1) + \log 3 + \frac{n}{3e}.$$

Substitute back, we can obtain the desired result. $\qquad\square$

# E  EXPERIMENTAL DETAILS

## E.1  ALGORITHM SUMMARY

Our algorithm is easily built on SAC. The only difference we make is we decouple the critic network into a representation network $\phi(\cdot)$ and a linear layer $l(\cdot)$ on top of the representation. The representation network is governed by the model dynamics loss in SPEDE, and we train a linear layer to predict the $Q$-value as it lies in the linear space of the representation guaranteed by our analysis. We update the representation by a momentum factor and keep the policy update the same procedure as SAC.

## E.2  FULL EXPERIMENTS

Table 2: Performance of SPEDE on various MuJoCo control suite tasks. Our method achieve strong performance even comparing to pure empirical baselines. To be specific, in hard tasks like Humanoid-ET and Ant-ET, SPEDE outperforms the baselines significantly. Results with $*$ are directly adopted from MBBL (Wang et al., 2019). We also provide the SoTA model-free RL method SAC as a reference.

|            | Swimmer   | Ant-ET        | Hopper-ET     | Pendulum   |
|------------|-----------|---------------|---------------|------------|
| ME-TRPO*   | 30.1±9.7  | 42.6±21.1     | 4.9±4.0       | 177.3±1.9  |
| PETS-RS*   | 42.1±20.2 | 130.0±148.1   | 205.8±36.5    | 167.9±35.8 |
| PETS-CEM*  | 22.1±25.2 | 81.6±145.8    | 129.3±36.0    | 167.4±53.0 |
| DeepSF     | 25.5±13.5 | 768.1±44.1    | 548.9±253.3   | 168.6±5.1  |
| **SPEDE**  | 42.6±4.2  | 806.2±60.2    | 732.2±263.9   | 169.5±0.6  |
| SAC*       | 41.2±4.6  | 2012.7±571.3  | 1815.5±655.1  | 168.2±9.5  |

|            | Reacher   | Cartpole   | I-pendulum   | Walker-ET     |
|------------|-----------|------------|--------------|---------------|
| ME-TRPO*   | -13.4±5.2 | 160.1±69.1 | -126.2±86.6  | -9.5±4.6      |
| PETS-RS*   | -40.1±6.9 | 195.0±28.0 | -12.1±25.1   | -0.8±3.2      |
| PETS-CEM*  | -12.3±5.2 | 199.5±3.0  | -20.5±28.9   | -2.5±6.8      |
| DeepSF     | -16.8±3.6 | 194.5±5.8  | -0.2±0.3     | 165.6±127.9   |
| **SPEDE**  | -7.2±1.1  | 138.2±39.5 | 0.0±0.0      | 501.58±204.0  |
| SAC*       | -6.4±0.5  | 199.4±0.4  | -0.2±0.1     | 2216.4±678.7  |

|            | MountainCar | Acrobot    | SlimHumanoid-ET | Humanoid-ET   |
|------------|-------------|------------|-----------------|---------------|
| ME-TRPO*   | -42.5±26.6  | 68.1±6.7   | 76.1±8.8        | 776.8±62.9    |
| PETS-RS*   | -78.5±2.1   | -71.5±44.6 | 320.7±182.2     | 106.9±102.6   |
| PETS-CEM*  | -57.9±3.6   | 12.5±29.0  | 355.1±157.1     | 110.8±91.0    |
| DeepSF     | -17.0±23.4  | -74.4±3.2  | 533.8±154.9     | 241.1±116.6   |
| **SPEDE**  | 50.3±1.1    | -69.0±3.3  | 986.4±154.7     | 886.9±95.2    |
| SAC*       | 52.6±0.6    | -52.9±2.0  | 843.6±313.1     | 1794.4±458.3  |

## E.3  ABLATIONS

Our ablation experiments are trying to study an important design choice of the practical algorithm: the momentum used to update the critic function. We summarize the results in Table 3. We can see that using a small large momentum factor such as 0.999 shows better performance. This is intuitively understandable: large momentum factor slows down the update speed of the representation of the critic function and thus stabilize the training. Such phenomenon illustrates the importance of slowly update the representation.

Table 3: Ablation Suty of SPEDE on MuJoCo tasks. We see that a small momentum factor help stabilize the performance, especially in environments like Huamoid and Hopper-ET.

|             | Hopper-ET       | Ant-ET         | S-Humanoid-ET   | Humanoid-ET   |
|-------------|-----------------|----------------|-----------------|---------------|
| SPEDE-0.9   | 593.2±37.4      | **877.7±45.9** | 881.6±385.2     | 232.9±63.4    |
| SPEDE-0.99  | 305.9±13.4      | 707.9±51.1     | 629.3±106.9     | 818.1±130.6   |
| SPEDE-0.999 | **732.2±263.9** | 806.2±60.2     | **986.4±154.7** | **886.9±95.2** |

### E.4 COMPARISON TO LC3

We provide a comparison of empirical results with LC3 (Kakade et al., 2020), which is also an algorithm with rigorous theoretical guarantees. Despite the major difference that we are learning the representation while LC3 assumes a given feature, the performance of SPEDE is much better than LC3 in tasks like Mountain Car and Hopper.

Table 4: Comparison of SPEDE with LC3 on MuJoCo tasks. LC3 only achieves good performance on relatively easy tasks like Reacher. However, their performance on Hopper and Mountain-Car is much worse than SPEDE.

|  | Reacher | MountainCar | Hopper |
|---|---|---|---|
| SPEDE | -7.2±1.1 | **50.3±1.1** | **732.2±263.9** |
| LC3 | **-4.1±1.6** | 27.3±8.1 | -1016.5±607.4 |

### E.5 PERFORMANCE CURVES

We provide an additional performance curve including ME-TRPO in Figure 2 for a reference.

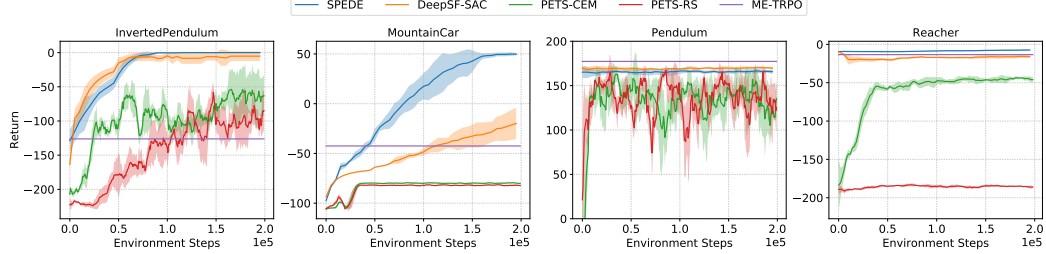

Figure 2: **Experiments on MuJoCo:** We show curves of the return versus the training steps for SPEDE and model-based RL baselines. We also include the final performance of ME-TRPO from (Wang et al., 2019) for reference.

### E.6 HYPERPARAMETERS

We conclude the hyperparameter we use in our experiments in the following.

Table 5: Hyperparameters used for SPEDE in all the environments in MuJoCo.

|  | Hyperparameter Value |
|---|---|
| Actor lr | 0.0003 |
| Model lr | 0.0001 |
| Actor Network Size | (1024, 1024, 1024) |
| Fourier Feature Size | 1024 |
| Discount | 0.99 |
| Target Update Tau | 0.005 |
| Model Update Tau | 0.001 |
| Batch Size | 256 |

