# OpenReview forum: "A Free Lunch from the Noise: Provable and Practical Exploration for Representation Learning"
_ICLR.cc/2022/Conference — ICLR 2022 Submitted_

### Official Review · Reviewer_8eEv · 2021-10-25

**Correctness:** 3
**Technical Novelty And Significance:** 2
**Empirical Novelty And Significance:** 2
**Recommendation:** 3
**Confidence:** 4

**Main Review:**

# Clarity and writing #

My first major comment is about clarity and writing. This paper emphasizes that the algorithm is doing representation learning. However, I believe that model-based learning might be more suitable for this paper. One major difference between the current paper and the prior representation learning/reward-free learning work is that the algorithm in this paper can only handle a single reward while previous model-based or model-free works can learn a representation that tackles multitasks (multiple rewards). I think such differences should be discussed in the paper. Since the algorithm can only a handle single reward, I do not see the significance and advantage of learning the "representation" (model). It belongs to the model-based algorithm where the TS agent tries to learn a model to address the reward-aware RL problem.

Moreover, some claims are not accurate. It is mentioned that “These algorithms learn a uniformly accurate model through a reward-free exploration, upon which decouple the learning from the exploration”, but Modi et al., 2021 is a model-free algorithm and does not try to learn a model.

Modi et al., 2021 first proposes to solve min-max-min optimization instead of min-max-min-max optimization as one operator has a closed-form solution. The min-max-min optimization is furthered solved in a more computationally efficient way by considering squared loss minimization and saddle-point problem.

For the theoretical part, the assumptions are not written clearly. For example, I believe equation (5) is a crucial assumption, but it is not discussed explicitly in Section 4.1.

The regret bound is about the Bayesian regret, and such guarantee is much weaker than the frequentist regret/worst case regret. The related discussion about the Bayesian regret and the definition of Bayesian regret E_{Pf} are missing.

I feel the title "a free lunch from the noise" oversells the paper. Indeed, it is a restricted assumption and the theoretical guarantee is obtained under such strong assumption.

# Technical analysis and algorithmic framework #
My second comment is about technical and algorithmic novelty. The algorithm is the standard model-based TS algorithm. Intuitively, given the model class, TS algorithm will identify the true model given a large amount of data. In addition, the authors make a strong assumption on the transition dynamics (eq 5) and the analysis is directly adapted from prior work (Russo & Van Roy (2013; 2014); Osband & Van Roy (2014)). I feel the technical contribution is rather limited.

# Experiments #
I think the most significant contribution of this paper is the experimental results. However, I'm not confident in evaluating the empirical results, and I feel there are a bunch of related model-based algorithms that achieve good empirical performance.

In the experimental part, it’s unclear how the algorithm is implemented. How do you choose the function class F and the prior distribution p(f)?

**Summary Of The Paper:**

This paper studies the single reward episodic MDP problem with the model-based TS algorithm. Assuming the given model class satisfies realizability, regularization property, bounded Eluder dimension, low covering number, and the true dynamics is the stochastic control model under gaussian noise transition (eq 5), the authors show the polynomial Bayesian regret guarantee. In addition, experiments on OPENAI MuJoCo is conducted.

**Summary Of The Review:**

I believe the clarity of the writing can be improved. It might not be a good idea to emphasize the algorithm is conducting representation learning and using the model-based learning might be more suitable.

The theoretical contribution is rather limited, and most significant contribution seems to be on the empirical side (I have to say I'm not familiar with the empirical work). The authors make a strong assumption on the true model. The algorithm and the analysis seems to be directly adapted from prior work, e.g., Russo & Van Roy (2013; 2014); Osband & Van Roy (2014).

It seems that there is a gap between the theoretical part and the experiment and it’s unclear how the experiment is conducted.

---

> ### Author Response · Authors · 2021-11-12
> **Discussion with Reviewer 8eEv**
>
> We thank the reviewer for the comments. However, we would like to kindly disagree with most of the arguments raised by the reviewer. We illustrate our idea below:
>
> * **On the Definition of Reinforcement Learning and Importance of Reinforcement Learning:** We want to briefly introduce the definition of representation learning in RL. As [1] pointed out, the linear transition dynamic is necessary for achieving small error on the planning. However, in [1], the authors assume the feature map is known and given to the learner, which does not hold in practice, as we discussed in Section 2. Therefore, any algorithms working on finding such feature map can be categorized as a representation learning in RL, and reward-aware/reward-free are just two different settings for representation learning. We would like to kindly argue that, even for the simple planning problem, that the exact reward and transition are given to the planner, it is still important to perform representation learning. Otherwise, there is no way to perform the planning efficiently and accurately. We believe our key observation helps filling the gap that is largely omitted by the community.
>
> * **Discussion about [3]:** Indeed, [3] is a model-free algorithm. However, we would like to emphasize the necessity of estimating the feature or the model uniformly accurate, under linear MDP assumption, to make sure the feature can be used with different rewards, no matter it is model-based or model-free. We have eliminated such ambiguity in the updated version.
>
> * **About Equation (5):** This is our model formulation which is widely used in both theoretical and empirical papers in literature cited here (please see the "On the Gaussian Noise" part in the General Response for a detailed explanation). Our discussion always builds upon our model definition, which is clearly explained with intensive discussion in Sec. 3.1.
>
> * **Bayesian Regre:** Please refer to the "TS v.s. UCB" part in the general response.
>
> * **Technical Novelty:** We compared our results with [4] and the discussion is in Section 4.2. We believe it is not straightforward to extend the results in [4] to show our case, as their results include a Lipschitz constant for the value function, while such dependency has been removed in our analysis. This will make our analysis more broadly applicable in practice, including the MuJoCo dynamics with collisions and neural dynamics with relu neurons.
>
> * **Practical Implementation and Empirical Comparison:**
>
>     As we emphasized in our maintext, we are targeting on seeking the practical and provable algorithm for representation learning in RL. We believe the proposed algorithm is the first algorithm satisfying such requirements. We demonstrate the superior of the proposed algorithm both theoretically and empirically.
>
>     We perform experiments of our paper on practical complex environments MuJoCo, which is a commonly used benchmark for many empirical model-based RL algorithms [5, 6, 7, 8]. We would also like to kindly remind the reviewer that the baselines we are comparing to are strong empirical algorithms without theoretical guarantees and there are very few theoretical RL methods can achieve comparable performance. We test SPEDE extensively on these tasks and it manage to achieve very competitive performance.
>
>     This reflects our effort on bridging the gap between theoretical and empirical side of RL. We believe this contribution should be appreciated when evaluating our paper.
>
> We believe our work reveal new directions on the representation learning in reinforcement learning, and provide both provable efficient and practical algorithms for a broad class of problem instances. Empirical results also show the effectiveness of our algorithm. We hope you can re-evaluate our paper carefully. Thanks in advance!

---

> ### Author Response · Authors · 2021-11-12
> **Reference for the Discussion with Reviewer 8eEv**
>
> [1] Jin, Chi, Zhuoran Yang, Zhaoran Wang, and Michael I. Jordan. Provably efficient reinforcement learning with linear function approximation. COLT 2020.
>
> [2] Alekh Agarwal, Sham Kakade, Akshay Krishnamurthy, and Wen Sun. Flambe: Structural complexity and representation learning of low rank mdps. NeurIPS 2020.
>
> [3] Aditya Modi, Jinglin Chen, Akshay Krishnamurthy, Nan Jiang, and Alekh Agarwal. Model-free representation learning and exploration in low-rank mdps.
>
> [4] Ian Osband and Benjamin Van Roy. Model-based reinforcement learning and the eluder dimension. NeurIPS 2014
>
> [5] Sham Kakade, Akshay Krishnamurthy, Kendall Lowrey, Motoya Ohnishi, Wen Sun. Information Theoretic Regret Bounds for Online Nonlinear Control. NeurIPS 2020.
>
> [6] Chua, Kurtland, Roberto Calandra, Rowan McAllister, and Sergey Levine. Deep Reinforcement Learning in a Handful of Trials using Probabilistic Dynamics Models. NeurIPS 2018
>
> [7] Kurutach, Thanard, Ignasi Clavera, Yan Duan, Aviv Tamar, and Pieter Abbeel. Model-Ensemble Trust-Region Policy Optimization. ICLR 2018
>
> [8] Ignasi Clavera, Jonas Rothfuss, John Schulman, Yasuhiro Fujita, Tamim Asfour, and Pieter Abbeel. Model-based reinforcement learning via meta-policy optimization. CoRL 2018,
>
> [9] Tingwu Wang, Xuchan Bao, Ignasi Clavera, Jerrick Hoang, Yeming Wen, Eric Langlois, Shunshi Zhang, Guodong Zhang, Pieter Abbeel, and Jimmy Ba. Benchmarking model-based reinforcement learning.

---

### Official Review · Reviewer_e9su · 2021-11-02

**Correctness:** 4
**Technical Novelty And Significance:** 3
**Empirical Novelty And Significance:** 3
**Recommendation:** 5
**Confidence:** 3

**Main Review:**

**Strengths**

I think overall this paper considers an interesting problem (learning representations) that is of interest to the community, given that a major part of the RL theory literature focuses on learning with a known linear representation or structured function class, and much less is known when it comes to learning the representation.

It is a plus that this paper contains empirical results too on real RL benchmarks, which I do appreciate as for a paper with a bulk part being theoretical contributions. The experimental results seem mostly convincing to me.

**Weaknesses**

One of my main concerns about the theory part is that this theory may be a somewhat cute but very specific consequence of the Gaussian noise assumption. It is kind of hard to tell how generalizable this result is (or how significant it is as a representation learning result), as the Gaussian structure (density of the noise admits this very form) is critically used instead of just for some concentration purposes. Though, I should remark from a more practical perspective I’m not that worried about using it for real-world problems, where it is hard to justify most structural assumptions anyway.

Also, from the current paper’s presentations, I find it quite hard to situate the current contributions in the context of representation learning theory in RL, and compare with related work. As concrete examples,

--- Section 3 (before 3.1) makes caveats of two prior techniques for representation learning: MLE, and policy cover technique. These techniques are discussed quite vaguely and not presented in math. Consequently I couldn’t find the key observation in 3.1 very motivated (since I do not see what the prior methods exactly are).

--- The real problem setting considered in this paper appears only in equation (5) and is not emphasized as a problem setting. I suggest maybe Section 3 could begin with talking about this setting, then talk about the existing approaches (with some math that at least hints what those methods are and why the authors think they are not sufficient), and then come to Section 3.1.

--- The authors discussed Theorem 5 in the context when $\mathcal{F}$ is a linear class; in this case I believe the model is *not* equivalent to linear MDP, could the authors comment more on what the difference is? Also, how does the present result compare with Agarwal et al. (2020) and Modi et al. (2021)?

--- Theorem 5 uses a Thompson sampling algorithm but the proof sketch mentions connection to a certain UCB algorithm. Are there reasons for not using UCB directly in Algorithm 1?

--- Could the authors discuss a bit more in detail how the proof compares with the standard Eluder Dimension proof (e.g. of Wang et al.)? Currently the proof sketch does not say much where the bounded Eluder dimension is used, and how is the application different from the existing Eluder dimension proof.

--- (This one is about experiments) When deployed into practice, the main difference between SPEDE and existing model-based algorithms is just that the parametrization (3) and (4) is used? (aside from SGLD as for approximate Bayesian inference + Gaussian policy parametrization, as the authors mentioned).

With the above questions in mind, I am left confused about the concrete novelty in the proof techniques or problem settings, and at most left with the impression that the result may be new but not sure how exactly it compares with prior work.

**Other comments**

--- The “practical issues in implementing the proposed SPEDE”: maybe this paragraph can form a standalone section after Section 4, since logically, the practical implementation tricks are not used in the theoretical results?

--- “expected regret” I believe it is usually called the Bayes regret in the thompson sampling literature?

--- Table 1, SPEDE-REG on MountainCar, the number should not be bolded?


**Summary Of The Paper:**

This paper studies the problem of learning representations for RL. On the theory side, the paper considers the setting where the state-transition is a nonlinear function of the past state-action plus additive noise, and develops a no-regret algorithm. On the empirical side, the paper shows that an adaptation of this algorithm on real-world RL tasks could perform better than existing model-based algorithms.

**Summary Of The Review:**

Overall, I think this paper makes interesting contributions on both the theoretical and empirical end of representation learning within RL. However, significant work needs to be done in order to clarify its problem setting, results, and position within related work.

---

> ### Author Response · Authors · 2021-11-12
> **Discussion with Reviewer e9su (1/2)**
>
> We thank the reviewer for the detailed comments and positive feedback on our effort to narrow the gap between theory and practice. We address the specific questions below:
>
> * **Gaussian Noise Assumption:** Please refer to the "On the Gaussian Noise" part in the general response.
> * **Prior Arts of Conditional MLE and Policy Cover:** Sorry for the confusion we made here due to the page limit. Generally, the previous representation learning algorithms take the following form:
> ** Collect data with the policies in the current policy cover set and append the data to the dataset.
> ** Perform representation learning based on the collected data.
> ** Planning on the learned representation to obtain the new policy cover set.
>
>   The policy cover set is carefully designed so that the union of the polices can be proved to cover all of the possible directions of the state representation with non-trivial probability (to make sure the predicted model can be generalized to all of the possible states).
>
>   In FLAMBE [1], assume we are given two representation function classes $\Phi$ and $\Upsilon$, one need to perform the representation learning via solving the following conditional MLE:
>   $$        \max_{\phi\in\Phi, \mu\in\Upsilon}\sum_{s, a, s^\prime} \log \phi(s, a)^\top \mu(s^\prime), \quad \sum_{s^\prime} \phi(s, a) \mu(s^\prime) = 1, \forall (s, a).
>   $$
>   As we need to guarantee all of the marginal distributions are valid, there are potentially infinite number of constraints, which is *computationally intractable* in general.
>
>   Moreover, as the representation learning procedure introduces additional error, the planning result with the estimated representation can be inaccurate. So in most of the prior arts, their planning steps generally return large numbers of policies to ensure that executing these policies can reach all of the states of interest, which is not practical in practice.
>
>   Instead, thanks to the noise structure, we do not really need to explicitly perform representation learning. We just need to estimate the dynamic function $f$, and obtain the desired representation with the structure of Gaussian noise for free. Since we just need to estimate the dynamic function, we can use standard OFU style algorithm to encourage the exploration, which also avoids the usage of policy cover for the exploration.
>
>     We hope such clarification can help you understand the difference between our algorithm and prior theoretical work, and position the proposed SPEDE in a better place: the only practical and provable representation learning for RL.
>
> * **On the Linear MDP Rerepsentation Learning:** Sorry for the confusion here. Our Theorem 5 holds even $\mathcal{F}$ is nonlinear.
>
>     The linear MDP [4] is defined below since the transition is linearly decomposed with known $\phi(s, a)$ (Equation (1) in the paper):
>
>     $$
>     T(s^\prime|s, a) = \langle \phi(s, a), \mu(s^\prime)\rangle.
>     $$
>     But linear $\mathcal{F}$ only has implications on the dynamic function $f$, i.e., $s' = As + Ba + \epsilon$, so linear $\mathcal{F}$ in fact does not directly imply the problem we consider is a linear MDP.
>
>     We instead, observe that we always have a linear MDP formulation for our considered model $s^\prime = f(s, a) + \epsilon$ when $\epsilon$ is isotropic Gaussian, with the help of (3) and (4) in the paper (also see (6)). This property in fact does not depend on the property of $f$ and $\mathcal{F}$, as we only use the structure of the noise, which removes potential constraints on the function class (e.g. linear or non-linear), and provides an efficient and accurate way to perform the planning with a given $f$, which eventually makes our algorithm perform well in practice.
>
> * **Compared with [1] and [2]** We would like to first sincerely remark, that our algorithm is *practical and implementable*, while their algorithms are not. [1] involves a conditional MLE that we have discussed above, and [2] involves a *min-max-min* optimization problem, which is hard to solve in general as well.
>
>     Furthermore, our results are not directly comparable with [1] and [2]. We consider one infinite dimensional linear MDP with special structure from the noise, while they consider finite dimensional linear MDP without special structures. Hence we obtain a strong regret bound with the benefit from the noise structure, which can guarantee we do not make many errors during the *whole learning procedure*, while they obtain a PAC bound, that only bounds the number of samples their algorithms need without guarantee on the errors they have made during the collection of samples.

---

> ### Author Response · Authors · 2021-11-12
> **Discussion with Reviewer e9su (2/2)**
>
> * **UCB v.s. TS:** Please refer to the "TS v.s. UCB" part in the general response.
>
> * **Compared with [3] and the usage of Bounded Eluder Dimension:** In [3], the authors consider the value function approximation with general function class, while we consider the dynamic function approximation, which already falls into different categories. Moreover, the algorithm proposed in [3] is in general not implementable, as it involved a constrained optimization that is generally intractable. Our algorithm instead, is implementable and enjoy strong empirical performance, as we have shown in the experiments.
>
>     The bounded Eluder dimension assumption is used in the Lemma 17, that bound the summation of width square $\sum_{k\in[K]}\sum_{h=0}^{H-1}\\|f^*(s_h^k, a_h^k) - \tilde{f}_k(s_h^k, a_h^k)\\|_2^2$. The bounded Eluder dimension guarantees that after some trails, we can estimate $f$ accurately, which eventually lead to an upper bound on this term.
>
> * **Difference in the Experiments:** Most of the empirical model-based RL algorithms [5, 6, 7, 8] does not utilize the linear MDP formulation, therefore, facing a difficult highly nonconvex optimization. These algorithms utilize different adhoc planning algorithms, e.g., CEM and random shooting [5]. Thus they will introduce extra approximation error and cannot guarantee to find the near-optimal policy, and hence degrade the performance. While the planning in our algorithm is much easier as the optimal $Q$-function lies in the linear space of the representation and this brings both theoretical and empirical benefit.
>
> * **Expected Regret/Bayesian Regret:** Yes, they generally refer to the same term, and different researchers have different preferences on its name.
>
> We hope our response can address your concerns. Please feel free to reach out to us if there are still some remaining concerns. And if there are no other issues, we hope you can adjust your score accordingly. Thanks in advance!

---

> ### Author Response · Authors · 2021-11-12
> **Reference for the Discussion with Reviewer e9su**
>
> [1] Alekh Agarwal, Sham Kakade, Akshay Krishnamurthy, and Wen Sun. Flambe: Structural complexity and representation learning of low rank mdps. NeurIPS 2020.
>
> [2] Aditya Modi, Jinglin Chen, Akshay Krishnamurthy, Nan Jiang, and Alekh Agarwal. Model-free representation learning and exploration in low-rank mdps.
>
> [3] Ruosong Wang, Russ R Salakhutdinov, and Lin Yang. Reinforcement learning with general value function approximation: Provably efficient approach via bounded eluder dimension. NeurIPS 2020
>
> [4] Jin, Chi, Zhuoran Yang, Zhaoran Wang, and Michael I. Jordan. Provably efficient reinforcement learning with linear function approximation. COLT 2020.
>
> [5] Chua, Kurtland, Roberto Calandra, Rowan McAllister, and Sergey Levine. Deep Reinforcement Learning in a Handful of Trials using Probabilistic Dynamics Models. NeurIPS 2018
>
> [6] Kurutach, Thanard, Ignasi Clavera, Yan Duan, Aviv Tamar, and Pieter Abbeel. Model-Ensemble Trust-Region Policy Optimization. ICLR 2018
>
> [7] Ignasi Clavera, Jonas Rothfuss, John Schulman, Yasuhiro Fujita, Tamim Asfour, and Pieter Abbeel. Model-based reinforcement learning via meta-policy optimization. CoRL 2018,
>
> [8] Tingwu Wang, Xuchan Bao, Ignasi Clavera, Jerrick Hoang, Yeming Wen, Eric Langlois, Shunshi Zhang, Guodong Zhang, Pieter Abbeel, and Jimmy Ba. Benchmarking model-based reinforcement learning.

---

### Official Review · Reviewer_azcj · 2021-11-02

**Correctness:** 4
**Technical Novelty And Significance:** 3
**Empirical Novelty And Significance:** 3
**Recommendation:** 8
**Confidence:** 3

**Details Of Ethics Concerns:**

No.

**Main Review:**

Strength. This paper extended the model in LC3[1] to kernel setting, and it also used Lemma 16 to remove the dependability on the global Lipschitz constant. It also conducted experiments to show that its algorithm is practical and efficient.

Weakness. After going through the proof, I notice that the regret bound is proportion to $\sigma^{-1}$, which means that the bound will be bigger when the noise level decrease and should be emphasized in Theorem 5. Here $\sigma$ is the noise level.

Although there are some details in Section 3.2, there should be more detail about the posterior sampling in the experiments section since this paper claimed to provide a practical algorithm and this part is not trivial.

[1]. Sham Kakade, Akshay Krishnamurthy, Kendall Lowrey, Motoya Ohnishi, and Wen Sun. Information-theoretic regret bounds for online nonlinear control. arXiv preprint arXiv:2006.12466,
2020.

**Summary Of The Paper:**

This paper proposed a practical exploration algorithm for finite-horizon RL problems and provided a theoretical guarantee for its algorithm when the transition kernel of the RL problem can be encoded with RKHS. It also conducted experiments to validate its algorithm.

**Summary Of The Review:**

This paper is significant, in the sense that it extends LC3[1] and provides a theoretical guarantee for a setting that is more complicated than [1]. It also uses a novel technique to utilize the noise in transition to remove the dependability on the global Lipschitz constant. Although the dependency on noise might have room for improvement, I think it is a good paper and could be accepted.

---

> ### Author Response · Authors · 2021-11-12
> **Discussion with Reviewer azcj**
>
> We thank the reviewer for the positive feedback. We address the issue below.
>
> * **Dependency of $\sigma^{-1}$:** Thanks for pointing it out! Indeed this is an artifact introduced by our proof strategy. The simulation lemma (Lemma 14) works well when the error $f(s, a) - \hat{f}(s, a)$ is small, but at the beginning of the learning process we cannot control the error of $f(s, a) - \hat{f}(s, a)$, and that's the reason we have an term $4C^2 H\mathrm{dim}_{E}(\mathcal{F}, T^{-1/2})$ in Lemma 18, and make our final regret bound scales with $\sigma^{-1}$. Notice that $\beta_K$ in equation (11) is proportion to $\sigma^2$ by setting $\alpha =\sigma^2 T^{-1/2}$, so the last term in Lemma 18 can absorb the $\sigma^{-2}$ in Lemma 14, which enables us to use the following alternative proof strategy:
> ** We can change the $\epsilon$ used in Lemma 18 to $\sigma^2 T^{-1/2}$, which will slightly change the eluder dimension, and make the constant $1$ in the upper bound introduced in Lemma 18 become $\sigma^2$.
> ** For the episodes involve the $Hdim_{E}(\mathcal{F}, \sigma^2 T^{-1/2})$ largest errors $w_{\mathcal{F_k}(s_h^k, a_h^k)}$ defined in Lemma 18, we just bound the regret for episode $k$ as $H$, and hence incur at most $H^2 dim_{E}(\mathcal{F}, \sigma^2 T^{-1/2})$ regret.
> ** For the episodes do not involve the the largest $Hdim_{E}(\mathcal{F}, \sigma^2 T^{-1/2})$ error $w_{\mathcal{F_k}(s_h^k, a_h^k)}$, we apply Lemma 14 and Lemma 18 to obtain the regret upper bound.
>
> Combine these steps, we can obtain a regret bound of $\sqrt{H^2 T (\frac{8\beta_K}{\sigma^2} + 1)dim_{E}(\mathcal{F}, \sigma^2 T^{-1/2})(1 + \log T)} + 0.5 H + H^2 \mathrm{dim}_{E}(\mathcal{F}, \sigma^2 T^{-1/2})$. As $\beta_K$ can be proportion to $\sigma^2$, our regret bound now does not scale with $\sigma^{-1}$. We have discussed this issue in Appendix C in the revised version. Note that this is only a proof strategy, and we do not need to make any change to the algorithm.

---

### Official Review · Reviewer_amjV · 2021-11-09

**Correctness:** 3
**Technical Novelty And Significance:** 2
**Empirical Novelty And Significance:** 3
**Recommendation:** 5
**Confidence:** 3

**Main Review:**

The paper studies an important problem in reinforcement learning, and shows some interesting observations. Here are some of my comments and questions.
1. In my opinion, this paper is more like one on model-based learning since Equation (5) is the main structure studied in the paper. It is remarked that "similar observations also hold for large amounts of distribution". I am wondering whether the theoretical results could be straightforwardly extended for other distributions other than Gaussian distribution.
2. I think a more detailed comparison to [1] in terms of technical novelty might be necessary. In my opinion, [1] focuses on frequentist regret bound, and thus proposes a UCB-like algorithm that requires an oracle. For computational efficiency, [1] also proposes a Thompson sampling (TS)-like algorithm that does not need such an oracle. In my opinion, it might be straightforward to show the Bayesian regret bound for this TS-like algorithm using the results in [1] and the standard and well-known results in [2]. In terms of proving Bayesian regret bound, could the authors elaborate more on the technical contributions beyond the existing literature?
3. In terms of the transition dynamics, I think it would be beneficial to the readers if the authors could say more on the dynamics that covered by the paper but not covered by [1]. In my opinion, providing some (simple) examples could be helpful.
4. Could the authors elaborate more on why the proposed algorithm decouples the exploration and representation learning since I think there is still longitudinal coupling across time-steps? Is this decoupling similar to that of Neural Linear algorithm in [3]?


[1] Sham Kakade, Akshay Krishnamurthy, Kendall Lowrey, Motoya Ohnishi, Wen Sun. Information Theoretic Regret Bounds for Online Nonlinear Control. NeurIPS 2020.
[2] Daniel Russo and Benjamin Van Roy. Learning to Optimize via Posterior Sampling, Mathematics of Operations Research, 2014.
[3] Carlos Riquelme, George Tucker, Jasper Snoek, Deep Bayesian Bandits Showdown: An Empirical Comparison of Bayesian Deep Networks for Thompson Sampling, ICLR 2018.

**Summary Of The Paper:**

This paper studies representation learning in the context of reinforcement learning, and observes that under some noise assumption, the linear spectral feature of corresponding Markov transition operator can be obtained in closed-form. Then the paper proposes the so-called SPEDE algorithm that enjoys good theoretical guarantee and empirical results.

**Summary Of The Review:**

I think this is an interesting paper, and I would like to understand better the technical novelty.

---

> ### Author Response · Authors · 2021-11-12
> **Discussion with Reviewer amjV**
>
> We thank the reviewer for the interests on the technical details. We address the main concern in the following:
>
> * **Extension to Other Distribution:** See ”On the Gaussian Noise” part in the general response.
>
> * **Technical Contributions:** Please refer to the ”TS v.s. UCB” part in the General Response for a discussion on frequentist regret and Bayesian Regret. We want to remark that, different from [1], we do not restrict the dynamics to be linear in certain feature map. This indeed improve the performance in our experiments. We provide the regret bound with the function class complexity defined in [2]. However, in [2] the authors only considered the bandit case, while we focus on the reinforcement learning scenarios, which involve additional technical difficulties. We want to reiterate, that when the function class only consists of linear function with respect to certain feature map, our results imply the regret bounds of [1], as we remarked in Section 4.2. Hence, our results is a strict generalization of the results from [1] that work without the strong assumption on the linear $f$. Compared with [3], our regret bound does not depend on the Lipschitz constant of the value function, which better suits the practical scenarios, as value functions in MuJoCo environments are not so smooth in general. We do believe our theoretical analysis extend the results from [1] and [3] and make the practical algorithm much more powerful.
>
> * **Dynamics Beyond the Case Covered by [1]:** Thanks for the suggestion! We would like to first notify, that [1] assumes the dynamics is linear w.r.t. the given feature map, which, as we have discussed in the remark of Section 4.2, is rare in practice. If we have no knowledge on the dynamics, we cannot design the feature map to make such assumption hold, so we want to go beyond known linear feature setting. For example, in the MuJoCo benchmark, if we have no prior knowledge on the representation (φ(x, u) in [1]), on which we can express the dynamics as a linear function, we can only take the raw states (x, u) from MuJoCo and make a linear function approximation on top of the non-informative prior transformation of (x, u). However, as the raw states of most of the MuJoCo tasks (e.g., Walker2D, Hopper, HalfCheetah) involves the observation on angle, angular velocity and torque of the agent, dynamics of these tasks cannot be expressed in simple linear form of the non-informative feature and linear function approximation on the non-informative feature cannot work in practice. This might be one of the reasons why [1] works pretty well in easy tasks like Reacher/InvertedPendulum, but fails on difficult tasks like Hopper.
>
> * **State-or-the-Art Performance that beats [1]:** We also want to sincerely reiterate that, our purpose is not providing theoretical guarantee for unrealistic algorithms, but designing a practical algorithm that have superior performance in practice while enjoy certain theoretical guarantees. As our algorithm allows for much more powerful function approximation, it beats the algorithm in [1] on the MuJoCo benchmark (see Appendix E.4 for the comparison) and approaches (or even beats) the state-of-the-art algorithms (as we shown in Section 5). We hope the reviewer can appreciate our effort on narrowing the gap between theory and practice.
>
> * **Coupled Representation Learning with Exploration:** Sorry for the confusion we made here. We want to express the idea, that our algorithm naturally handles the coupled exploration and representation learning in a systematic sequential way. Specifically, once we obtain the posterior of the model for exploration, the representation is obtained from the sampled model from the posterior via the noise structure. The exploration and representation extraction steps are algorithmically decoupled. In this sense, the exploration and representation are separated through the posterior of model. We already modified the corresponding part in the main text to make this more clear. We want to remark that [6] focused on the bandit setting without representation learning, which is not directly comparable to our work.
>
> If there are other concerns on our paper, please feel free to reach out to us. If our response properly address your concerns, we hope you can adjust your evaluation accordingly. Thanks in advance!

---

> ### Author Response · Authors · 2021-11-12
> **References for the Discussion with Reviewer amjV**
>
> [1] Sham Kakade, Akshay Krishnamurthy, Kendall Lowrey, Motoya Ohnishi, Wen Sun. Information The- oretic Regret Bounds for Online Nonlinear Control. NeurIPS 2020.
>
> [2] Daniel Russo and Benjamin Van Roy. Learning to Optimize via Posterior Sampling, Mathematics of Operations Research, 2014.
>
> [3] Ian Osband and Benjamin Van Roy. Model-based reinforcement learning and the eluder dimension. NeurIPS 2014
>
> [4] Alekh Agarwal, Sham Kakade, Akshay Krishnamurthy, and Wen Sun. Flambe: Structural complexity and representation learning of low rank mdps. NeurIPS 2020.
>
> [5] Aditya Modi, Jinglin Chen, Akshay Krishnamurthy, Nan Jiang, and Alekh Agarwal. Model-free repre- sentation learning and exploration in low-rank mdps.
>
> [6] Carlos Riquelme, George Tucker, Jasper Snoek, Deep Bayesian Bandits Showdown: An Empirical Comparison of Bayesian Deep Networks for Thompson Sampling, ICLR 2018.

---

### Public Comment · ~Philip_Ball1 · 2021-11-10
**Nice work!**

Hi there, thank you for your very interesting paper! We believe that there is uncited related work that relies on a similar principle, which is using noise to induce optimistic planning to facilitate exploration in the online MDP. In [1], instead of sampling a representation posterior, we leverage a dynamics posterior from an ensemble of models, and use this noise to plan provably efficient policies using OFU principles. Thank you!

[1]  Towards Tractable Optimism in Model-Based Reinforcement Learning, Pacchiano et al., UAI 2021

---

> ### Author Response · Authors · 2021-11-12
> **Thanks for your interest!**
>
> Hi Philip. Thanks for providing the interesting work! However, we want to kindly remark that, we use the noise distribution in a totally different way. We utilize the structure of noise distribution and provide a representation for the $Q$ function that can be used for planning, while the referred paper can be viewed as a mechanism to efficiently obtain the optimistic model and the corresponding optimistic policy with extra stochasticity introduced. We will add a detailed discussion on the difference between our method and the referred paper. Thanks for reading our paper again!

---

### Author Response · Authors · 2021-11-12
**General Response**

We thank all of the reviewers for the detailed comments. We would like to reemphasize our core contribution to the community, and provide the feedback for several general concerns here.

* **Core Contribution:** We want to reiterate that, our main purpose, as we claimed in the introduction, is to provide **provable efficient and practical** algorithm, whose performance should be guaranteed by the theoretical justification and demonstrated by the empirical justification. There are some previous work considered much more general setting, like FLAMBE [1] and MOFFEL [2]. However, as we pointed out in Section 1.1, they are **not practical and only of theoretical interest**. We instead, propose an easy to implement algorithm, that focused on the real control problem and outperforms the state-of-the-art result on multiple MuJoCo benchmarks. We hope the reviewers can pay more attention on our algorithm's success on both empirical and theoretical aspects, which also demonstrates the effectiveness of our method.

* **On the Gaussian Noise:** We would like to remark that, Gaussian noise is a general assumption, that is widely used in both theory [3, 4, 5, 6, 7] and practice [8, 9, 10, 11]. Hence, we believe it is sufficient for our paper to focus on the Gaussian noise. It is indeed possible to generalize our result to other distributions, based on the following senses:
** As we have pointed out in Section 3.1, there are other distributions that are known to be factorized without further computation, e.g. the Laplace distribution.
** Our proof only leverages the Gaussian noise in the learnability of the function class (Lemma 13) and the simulation lemma (Lemma 14). The learnability of the function class can also be obtained by proper moment conditions on the noise, and it is possible to show different simulation lemmas with different noises, which we tend to leave as future work.

* **TS v.s. UCB:** We would like to emphasize that, we focus on the TS algorithm in the paper, as it is much easier to implement in practice. We also discuss the equivalent UCB algorithm in Appendix B. Note that, UCB algorithm needs to explicitly maintain the confidence set and perform constrained optimization in each episode, which is computationally intractable for general function classes. Furthermore, our proof is originated from the UCB algorithm and provides a frequentist regret upper bound for it (see the Theorem 12 in the revised version). We translate the frequentist regret upper bound for the UCB algorithm to the Bayesian regret upper bound for the TS algorithm following the same strategy of [12]. It works for any prior distribution, but at the expense of an bound only on the expected regret, that is weaker than the bound on the worst case regret we proved for the UCB algorithm.

* **Comparison to LC3 [7]:** We also provide a comparison of SPEDE with LC3 [7] in Appendix E.4. We would like to emphasize that LC3 only conducted experiments on relatively easy tasks in MuJoCo. The only relatively hard task they did is Hopper, where their performance is much worse than SPEDE. For other hard tasks like Walker, HalfCheetah, they did not manage to run it.

* **Comparison of SPEDE with LC3** (Higher is better)
| |Reacher|MountainCar|Hopper|
|--|--|--|--|
|SPEDE| -7.2$\pm$ 1.1| 50.3 $\pm$ 1.1| 732.2 $\pm$ 263.9|
|LC3| -4.1 $\pm$ 1.6| 27.3 $\pm$ 8.1|-1016.5 $\pm$ 607.4|

---

> ### Author Response · Authors · 2021-11-12
> **Reference in the General Response**
>
> [1] Alekh Agarwal, Sham Kakade, Akshay Krishnamurthy, and Wen Sun. Flambe: Structural complexity and representation learning of low rank mdps. NeurIPS 2020.
>
> [2] Aditya Modi, Jinglin Chen, Akshay Krishnamurthy, Nan Jiang, and Alekh Agarwal. Model-free representation learning and exploration in low-rank MDPs. ArXiv 2021.
>
> [3] Yasin Abbasi-Yadkori and Csaba Szepesvari. Regret bounds for the adaptive control of linear quadratic systems. COLT 2011
>
> [4] Horia Mania, Stephen Tu, and Benjamin Recht. Certainty equivalence is efficient for linear quadratic control. NeurIPS 2019
>
> [5] Horia Mania, Michael I Jordan, and Benjamin Recht. Active learning for nonlinear system identification with guarantees. 2020.
>
> [6] Max Simchowitz and Dylan Foster. Naive exploration is optimal for online LQR. ICML 2020
>
> [7] Sham Kakade, Akshay Krishnamurthy, Kendall Lowrey, Motoya Ohnishi, and Wen Sun. Information theoretic regret bounds for online nonlinear control. NeurIPS 2020.
>
> [8] Chua, Kurtland, Roberto Calandra, Rowan McAllister, and Sergey Levine. Deep Reinforcement Learning in a Handful of Trials using Probabilistic Dynamics Models. NeurIPS 2018
>
> [9] Kurutach, Thanard, Ignasi Clavera, Yan Duan, Aviv Tamar, and Pieter Abbeel. Model-Ensemble Trust-Region Policy Optimization. ICLR 2018
>
> [10] Ignasi Clavera, Jonas Rothfuss, John Schulman, Yasuhiro Fujita, Tamim Asfour, and Pieter Abbeel. Model-based reinforcement learning via meta-policy optimization. CoRL 2018,
>
> [11] Tingwu Wang, Xuchan Bao, Ignasi Clavera, Jerrick Hoang, Yeming Wen, Eric Langlois, Shunshi Zhang, Guodong Zhang, Pieter Abbeel, and Jimmy Ba. Benchmarking model-based reinforcement learning.
>
> [12] Ian Osband and Benjamin Van Roy. Model-based reinforcement learning and the eluder dimension. NeurIPS 2014

---

> ### Comment · Reviewer_amjV · 2021-11-13
> **Thompson sampling variant of LC3**
>
> Thanks a lot for the response. In my opinion, Kakade et al. [1] indeed proposes the Thompson sampling (TS) variant of LC3, which is practical and computationally efficient. In terms of provable efficiency, using the standard and well-known results in Russo and Van Roy [2], it could be straightforward to transform the frequentist regret of LC3 to the Bayesian regret of TS variant of LC3. In my opinion, to achieve better technical contribution, it would be great to show the frequentist regret of SPEDE. In terms of numerical results, is the comparison between the TS variant of LC3 and SPEDE? If not, I would like to see the empirical comparison between these two, which might show the advantage of SPEDE in a clearer way.
>
>
>
>
> [1] Sham Kakade, Akshay Krishnamurthy, Kendall Lowrey, Motoya Ohnishi, Wen Sun. Information Theoretic Regret Bounds for Online Nonlinear Control. NeurIPS 2020.
>
> [2] Daniel Russo and Benjamin Van Roy. Learning to Optimize via Posterior Sampling, Mathematics of Operations Research, 2014.

---

> > ### Author Response · Authors · 2021-11-14
> > **Thanks for your reply! We illustrate our idea below.**
> >
> > Thanks for your reply! We conclude our idea below:
> >
> > * **Frequentist Regret**:  Indeed, as we clearly stated in the "TS v.s. UCB part", we have both frequentist and Bayesian regret in the updated version. The frequentist regret for SPEDE is shown in Theorem 12. We would like to kindly reiterate that, as we discussed in Section 4.2, when the function class $\mathcal{F}$ is the linear function class considered in [1], our regret bounds directly imply the results in [1]. Hence, our results is a strictly generalization of [1], and [1] cannot directly imply our results.
> >
> > * **Comparison between SPEDE and LC3**:  The performance comparison listed in the table is between the TS variant of SPEDE and LC3. As we can see, SPEDE performs better than LC3, especially on the task Hopper with complicated dynamics. We emphasize SPEDE considers generally nonlinear dynamics, while LC3 only consider the linear dynamics with respect to certain feature map. If linear function on top of this feature map contains the real dynamic $f$ and we use this linear function class, SPEDE degenerates to LC3 with the same regret (as we discussed in Sec. 4.2). However, as we have discussed in the "Technical Contribution" and "Dynamics Beyond the Case Covered by [1]" part in the discussion with you, such linear function class is generally unknown, which constrains the performance of LC3, and we instead leverage much more powerful function approximation like neural nets, which significantly improve the performance on the environment with complicated dynamics.
> >
> > [1] Sham Kakade, Akshay Krishnamurthy, Kendall Lowrey, Motoya Ohnishi, and Wen Sun. Information theoretic regret bounds for online nonlinear control. NeurIPS 2020.

---

> > > ### Comment · Reviewer_amjV · 2021-11-14
> > > **frequentist regret for TS-SPEDE**
> > >
> > > Thanks for the reply! If my understanding is correct, the original SPEDE uses TS as a subroutine (call it TS-SPEDE), while it seems that Theorem 12 is for UCB-SPEDE? It would be a technical contribution beyond Kakade et al. [1] if the frequentist regret for TS-SPEDE is shown.

---

> > > > ### Author Response · Authors · 2021-11-14
> > > > **On the Frequentist Regret for TS**
> > > >
> > > > Thanks for your reply! However, we do believe that frequentist regret for TS algorithm is out of the scope of our paper. In fact, little is known about the frequentist regret for TS algorithm. There are some exceptions like [1, 2, 3], but they focused on the linear bandit setting or tabular MDP setting, that has certain closed-form for the posterior update, which helps the analysis. For general function class, to the best of our knowledge, we are not aware of any frequentist regret analysis on that.
> > > >
> > > > Furthermore, we would like to reiterate, that our purpose is providing practical algorithm that have good empirical performance with theoretical guarantee. We don't want to solve the open problem of frequentist regret bound of TS. Instead, we want to relax the constraints of linear function class in LC3 to further improve the empirical performance, and also provide theoretical results for our algorithm that include the results of LC3 as special case, which we believe is already a solid technical contribution to the potential audience. We hope the reviewer can focus on our effort on narrowing the gap between theory and practice.
> > > >
> > > > [1] Agrawal, Shipra, and Navin Goyal. "Thompson sampling for contextual bandits with linear payoffs." ICML 2013
> > > >
> > > > [2] Abeille, Marc, and Alessandro Lazaric. "Linear thompson sampling revisited." AISTATS 2017
> > > >
> > > > [3] Agrawal, Shipra, and Randy Jia. "Posterior sampling for reinforcement learning: worst-case regret bounds." NeurIPS 2017

---

### Decision · Program_Chairs · 2022-01-20

**Decision:**

Reject

**Comment:**

The AC summarizes the major strengths and weaknesses of the paper pointed out by the reviewers (with possible omissions, and additions by the AC)

Strengths:
1. The paper makes an important observation that the linear MDP assumptions can be met when the true dynamics has additive noise
2. Inspired by the theory, the paper proposes a new algorithm that empirical outperforms SAC. The success of the algorithm is very interesting (and surprising to some degree.)

Weaknesses
3. Most of the reviewers and the AC thinks the representation learning perspective is questionable. If one strongly believes that the $\phi, \mu$ in the linear MDP assumption should be interpreted as representations, then yes, this paper is about representation learning in RL and the representation learning is a free lunch. However, suppose one ignores the linear MDP perspective for the moment, and only looks at the modeling assumption $s' = f^*(s,a) +\epsilon$, then $f^*$ can only be interpreted as a "dynamics model" and has nothing to do with the term "representation" that is commonly used in practice. (representation means the penultimate layer of the neural nets typically in emprical RL.)  Moreover, in the theory part of the paper, the dynamics model is learned via a (standard) model-based approach---fitting $f(s,a)$ to $s'$---which also suggests that $f$ should be interpreted as a dynamics model instead of a representation. How to reconcile these two perspectives? The AC's own opinion is that this suggests we shouldn't blindly call the $\phi$ in the linear MDP formulation a representation in all scenarios. But regardless of AC's own opinion, I suspect that the paper needs to very explicitly discuss and clarify these discrepancies (instead of somewhat sweeping it under the rug and claiming the paper is about representation learning without a stronger justification.)

4. The sample efficiency depends on the Eluder dimension, which is only known to be polynomial for linear models. Recent works have shown that the Eluder dimension for even simple nonlinear models can be exponential. The analysis seems to be also quite related to previous analysis that uses the Eluder dimension. I think this fact limits the theoretical contribution of the paper.

5. There could be a better exposition of the empirical implementation in the paper. It appears that the implemented algorithm still has some major differences from the theoretical algorithms.

6. It's unclear if the paper should only compare with model-free algorithms. At least the theoretical algorithm fits $f(s,a)$ to $s'$ explicitly (in the definition of confidence region). Therefore, it does not seem to be quite fair to compare with model-free algorithms.

Given these considerations, and given that the majority of the reviewers express some concerns about various subsets of these concerns (3-6), the AC would recommend the authors revise the paper and resubmit to another top ML venue. The AC thinks that the paper contains really interesting and novel observations, but the interpretation of the observation might require more thoughts and clarification.